# TOKENIZING SINGLE-CHANNEL EEG WITH TIME-FREQUENCY MOTIF LEARNING

**Jathurshan Pradeepkumar**[1*] **Xihao Piao**[2], **Zheng Chen**[2,*] **Jimeng Sun**[1,*]
[1]University of Illinois Urbana-Champaign   [2]SANKEN, Osaka University
{jp65,jimeng}@illinois.edu,{park88,chenz}@sanken.osaka-u.ac.jp

## ABSTRACT

Foundation models are reshaping EEG analysis, yet an important problem of EEG tokenization remains a challenge. This paper presents `TFM-Tokenizer`, a novel tokenization framework that learns a vocabulary of time-frequency motifs from *single-channel* EEG signals and encodes them into discrete tokens. We propose a dual-path architecture with time–frequency masking to capture robust motif representations, and it is model-agnostic, supporting both lightweight transformers and existing foundation models for downstream tasks. Our study demonstrates three key benefits: *Accuracy:* Experiments on four diverse EEG benchmarks demonstrate consistent performance gains across both single- and multi-dataset pretraining settings, achieving up to $11\%$ improvement in Cohen's Kappa over strong baselines. *Generalization:* Moreover, as a plug-and-play component, it consistently boosts the performance of diverse foundation models, including BIOT and LaBraM. *Scalability:* By operating at the single-channel level rather than relying on the strict 10–20 EEG system, our method has the potential to be device-agnostic. Experiments on ear-EEG sleep staging, which differs from the pretraining data in signal format, channel configuration, recording device, and task, show that our tokenizer outperforms baselines by $14\%$. A comprehensive token analysis reveals strong class-discriminative, frequency-aware, and consistent structure, enabling improved representation quality and interpretability. Code is available at https://github.com/Jathurshan0330/TFM-Tokenizer.

## 1 INTRODUCTION

Foundation models have revolutionized how machines understand human language, leading to major breakthroughs in natural language processing (NLP) (OpenAI et al., 2024; DeepSeek-AI et al., 2025) and cross-modality tasks such as text-to-image generation (Bordes et al., 2024). Inspired by this success, researchers are now advancing a paradigm shift in electroencephalogram (EEG) analysis toward task-agnostic foundation models (Mohammadi Foumani et al., 2024; Yang et al., 2024; Jiang et al., 2024b; Wang et al., 2024a). By pretraining on massive, diverse EEG data corpora, these models learn universal representations that generalize well across various downstream tasks.

Despite substantial recent progress, an important open problem remains: *how to design an effective tokenization method for EEG signals.* Tokenization, a core component in NLP, transforms raw text into meaningful tokens, which reduces data complexity and introduces a helpful inductive bias in foundation models (Gastaldi et al., 2025). Typically, tokenization is performed by a learnable function that trains a vocabulary of tokens and statistics from a given corpus. However, existing EEG foundation models tokenize signals by directly segmenting continuous EEGs into short-duration tokens, without learning a vocabulary. They merely discretize EEG signals, failing to capture statistically grounded representations in a data-driven manner. LaBraM (Jiang et al., 2024b) proposes a neural tokenizer to learn data-driven tokens before pretraining. However, these tokens primarily serve as training objectives rather than as actual inputs for subsequent model training and are discarded during downstream inference, limiting their reusability. As a result, the foundation model is still trained on continuous segment-level embeddings, failing to fully leverage the benefits of tokenization, such as improving the quality of input representations. In this paper, we study a novel and critical problem of developing a principled EEG tokenization that seamlessly integrates with various foundation models and enhance downstream performance and generalization.

---

*corresponding authors

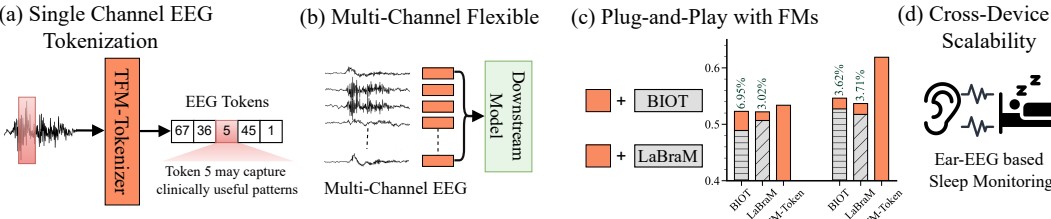

Figure 1: (a) Our TFM-Tokenizer converts single-channel EEG into discrete tokens by capturing time-frequency motifs. (b) It is adaptable to any different multi-channel settings, (c) can be integrated with existing foundation models to enhance their performance, and (d) enables cross-device scalability.

Various studies have shown that developing an effective tokenization is a non-trivial task in general, as it is influenced by multiple factors (Schmidt et al., 2024). In this paper, we recognize and focus on three key challenges of EEG tokenization. **1) Tokenization target:** real-world EEG recordings exhibit diverse formats due to varying devices, channel configurations, and recording lengths (Yang et al., 2024). We argue that tokenizers should be trained and operated at the *single-channel level* to learn channel-agnostic discrete tokens. This design enables flexible adaptation to multi-channel tasks and can generalize to non-standard EEG devices. In Section 4.4, we provide scalability experiments on ear-EEG settings. **2) Token resolution:** in NLP, tokenization can be defined at different resolutions (characters, subwords, words), each reflecting different assumptions about semantic granularity. However, EEG signals are characterized by diverse oscillatory (e.g., alpha, beta) (Pradeepkumar et al., 2024) and transient patterns (e.g., spikes) (Chen et al., 2022). Thus, effective tokens must represent such underlying *motifs* (Xu et al., 2023) that reflect distinct neural or physiological events. Motifs can be understood as short, recurring patterns in a time series that exhibit limited variability and often carry discriminative significance (Xu et al., 2023). However, these motifs are often distorted by noise, amplitude scaling, and temporal warping, making it challenging to design robust EEG tokenization methods. **3) Tokenization learning objective:** EEGs exhibit various temporal variations, manifested as a mixture of low- and high-frequency components that co-occur and are intermixed in complex ways. Relying solely on capturing time-based motifs into discrete tokens and expecting the model to implicitly infer spectral structure from raw signals risks overlooking important frequency information. We therefore argue that the tokenization learning objective should explicitly incorporate *time–frequency representations*, enabling the tokenizer to capture band-specific and cross-frequency patterns and to encode more meaningful neural motifs

To tackle these challenges, we propose `TFM-Tokenizer`, a novel EEG tokenization framework that captures time–frequency motifs from single-channel EEG signals and encodes them into distinct tokens. Specifically, **1) Tokenizing EEGs at single-channel:** We tokenize single-channel EEG signals into discrete token sequences akin to NLP models, which are then paired with a generic transformer to perform multi-channel modeling using these single-channel tokens. Our tokenizer is model-agnostic and can be paired with any downstream model. Our experiments confirmed that TFM-Tokenizer can seamlessly integrate with existing foundation models, and further improve their performance (see Figure 1). **2) Learning motif features as tokens:** We introduce a motif learning architecture that encodes time–frequency motifs into tokens through a dual-path encoding design. Capturing frequency-band characteristics or compositions is crucial for EEG analysis, and to model such dynamics, we designed a Localized Spectral Window Encoder, which isolates and aggregates information across frequency bands prior to fusion with temporal features. **3) Explicit time-frequency masking prediction:** this learning objective disentangles the entangled time–frequency representations, enabling the model to explicitly learn distinct frequency-specific patterns across time. By forcing the model to predict masked regions in both domains, it encourages the tokenizer to discover and encode meaningful neural motifs that are localized in time and frequency. Overall, our contributions are summarized as follows:

- **Formulating Single-Channel EEG Tokenization.** To our knowledge, we are the first to investigate the problem of learning a discrete token vocabulary that captures time–frequency motifs in *single-channel* EEG signals from a given corpus and directly utilizes them as inputs for downstream modeling.

- **Proposing Novel TFM-Token Framework.** We introduce a single-channel EEG tokenization framework that transforms EEG into a discrete token sequence via TFM-Tokenizer, which is then

used by a lightweight transformer model for cross-channel and downstream modeling. As shown in Figure 1c, TFM-Tokenizer integrates smoothly with existing models and consistently boosts performance, improving BIOT and LaBraM by approximately $4\%$ on TUEV dataset.

- **Broad Evaluation across Foundation Models and Devices.** Extensive experiments across four datasets show that our method outperforms strong baselines, achieving up to a $11\%$ gain over the baseline model on TUEV dataset. We also evaluate cross-device scalability on an ear-EEG sleep staging task, using electrodes outside the standard 10–20 EEG system, where our tokenizer outperforms baselines by $14\%$. Beyond performance, we comprehensively analyze token quality, including token consistency, class-specific uniqueness, and frequency learning analysis, validating that our learned tokens are informative and interpretable.

## 2 RELATED WORK

**EEG Foundation Models and Tokenization Methods.** Existing EEG foundation models can be categorized into decoding and encoder-based methods. Decoding-based methods focus on generative tasks like cross-modal translation (Duan et al., 2023; Liu et al., 2024; Wang et al., 2024c). In contrast, encoder-based methods focus on classification tasks and representation learning. Notable models include LaBraM (Jiang et al., 2024b), BIOT (Yang et al., 2024), BRANT (Zhang et al., 2024), and MMM (Yi et al., 2024). Our work aligns with this latter category, aiming to enhance input representations to improve classification performance and generalization across diverse foundation models. A parallel question is how to *tokenize* EEG signals. Existing methods primarily adopt segment-based continuous tokenization (Yang et al., 2024; Wang et al., 2024b; Zhang et al., 2024). Vector Quantized (VQ) tokenizers (Van Den Oord et al., 2017), which have been successful in tokenizing continuous images (Esser et al., 2020), have recently been adapted for EEG by LaBraM (Jiang et al., 2024b). However, in LaBraM, the tokenizer is not designed to represent EEG data and replace raw signals as inputs to foundation models; instead, it mainly serves as a training objective. In this paper, we propose a new tokenization framework for EEG signals that encodes inputs into discrete representations and provide a reusable interface for foundation models.

**EEG Motif Learning.** Motifs are short, recurring patterns with small variability in a time series and may hold predictive or discriminative value (Xu et al., 2023). In the EEG domain, motif learning has been studied in several prior works, including (Mueen et al., 2009; Barthélemy et al., 2013; Gips et al., 2017; Jas et al., 2017; Schäfer & Leser, 2022), though these approaches focus solely on the temporal domain. EEG motifs correspond to neurophysiological events such as oscillatory bursts or transient spikes, which are best characterized by joint temporal-spectral structure. Frequency-domain modeling is therefore essential, yet raw time-domain signals often entangle multiple spectral components. This can cause models to overemphasize dominant low-frequency rhythms while overlooking informative high-frequency details (Zhi-Qin John Xu et al., 2020; Piao et al., 2024). Such bias limits the ability to capture diverse EEG waveforms and degrades representation quality (Park & Kim, 2022). To the best of our knowledge, we are the first to propose methods to encode diverse, informative time–frequency motifs as discrete tokens.

## 3 METHODOLOGY

### 3.1 FRAMEWORK OVERVIEW AND FORWARD PROCESS

Our `TFM-Tokenizer` framework consists of two major phase, as shown in Figure 2:

1. **TFM-Tokenizer with Motif Learning.** The tokenizer is trained in a single-channel, unsupervised setting, capturing key motif features. We regard motifs as various waveforms that encode characteristic time–frequency patterns in EEGs. To represent these motifs, the tokenizer is composed of four components: (i) a Localized Spectral Window Encoder that extracts frequency patterns within short spectral windows, (ii) a Temporal Encoder that incorporates raw EEG context, (iii) a Temporal Transformer that models dependencies across windows, and (iv) a codebook quantizer that maps embeddings into a discrete vocabulary. Therefore, we train a motif-based vocabulary that transforms continuous EEGs into interpretable discrete tokens (Sec. 3.2).

2. **Downstream Transformer Model.** This phase serves as an example to illustrate *how a foundation model processes tokenized sequences for downstream tasks* such as classification. Raw

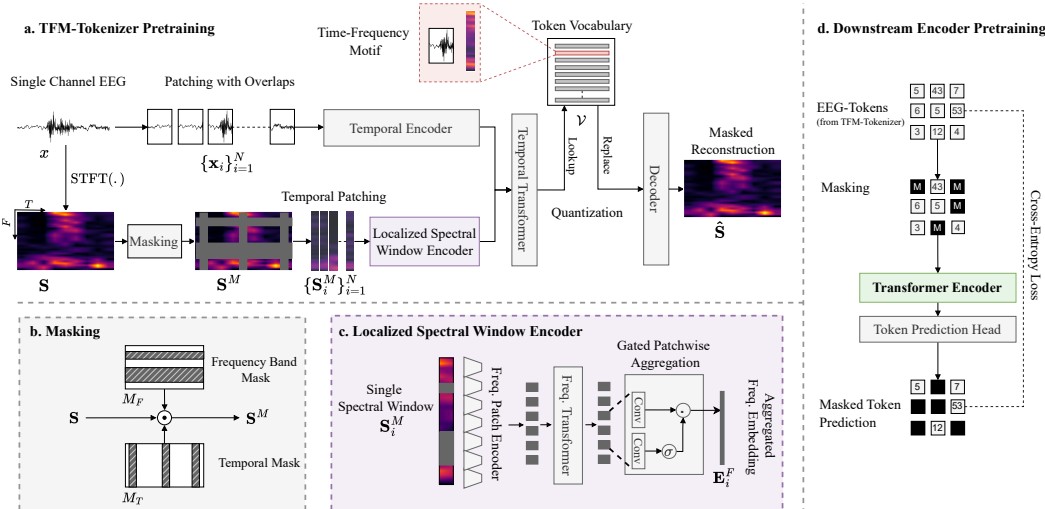

Figure 2: Overview of our framework. (a) `TFM-Tokenizer` Pretraining: Through dual-path encoding and masked prediction, learns to capture time-frequency motifs into discrete tokens. (b) Masking Strategy: A combination of frequency band masking and temporal masking is used for `TFM-Tokenizer` pretraining. (c) Localized Spectral Window Encoder: Processes individual spectral windows from $\mathbf{S}$, extracts frequency band information, and aggregates features across all bands into a single compact embedding per window. (d) Downstream Transformer Encoder Pretraining: Trains on learned EEG tokens using masked token prediction.

> EEGs are first passed through our pretrained tokenizer, where they are converted into discrete tokens that serve as inputs to foundation models. Since the tokenizer is model-agnostic, it can be paired with different backbone models. In our implementation, we adopt a lightweight Transformer (Vaswani et al., 2017) with linear attention (Katharopoulos et al., 2020), demonstrating that the tokenizer ($\sim$0.7M parameters) enables strong performance even with a compact model (Sec. 3.3).

Overall, we first pretrain the tokenizer to learn a discrete vocabulary of EEG motifs. The tokenizer is then frozen, and the downstream Transformer is pretrained with a masked token prediction objective. Finally, the downstream Transformer is fine-tuned on target EEG tasks such as classification.

## 3.2 SINGLE-CHANNEL TFM-TOKENIZER WITH MOTIF LEARNING

`TFM-Tokenizer` encodes EEGs into discrete motifs tokens through a dual-path frequency–time paradigm (Figure 2a). Given a multi-channel EEG $\mathbf{X} \in \mathbb{R}^{C \times T}$, we segment each channel signal $x$ into overlapping patches of length $L$ and hop size $H$, yielding $N = \lfloor (T-L)/H \rfloor + 1$ patches aligned with spectral windows $\{\mathbf{S}_i\}_{i=1}^N$. To define the pretraining task, masking is applied in both temporal and frequency domains (Figure 2b), where unmasked patches provide context and masked ones are reconstructed. Feature learning is performed as follows: each spectral window $\mathbf{S}_i$ is encoded by the Localized Spectral Window Encoder (Figure 2c) and fused with raw EEG patch features through a Temporal Encoder. A Temporal Transformer then integrates the time–frequency features, and the output embeddings are mapped into a learnable VQ vocabulary, producing motif tokens.

**Localized Spectral Window Encoder.** Capturing frequency-band characteristics is essential for EEG analysis, as the signals often exhibit oscillatory components (e.g., alpha, beta) with varying amplitudes and temporal dynamics. Unlike prior work that projects an entire spectral window through a single linear layer (Yang et al., 2024), we divide the window into patches along the frequency axis, allowing effective modeling of cross-frequency dependencies. This process consists of three steps.

- *Frequency Patch Encoder.* Given a set of spectral windows $\{\mathbf{S}_i\}_{i=1}^N$, we isolate and divide each spectral window $\mathbf{S}_i$ into $P$ non-overlapping patches $\{\mathbf{S}_{(i,p)}\}_{p=1}^P$, each spanning $\Delta f$ frequency bins such that $P.\Delta f = F$. We then project each frequency patch into a latent space: $e_{(i,p)} = \text{GroupNorm}\left(\text{GeLU}\left(\mathbf{W}_p \mathbf{S}_{(i,p)}\right)\right)$ where $\mathbf{W}_p \in \mathbb{R}^{D \times \Delta f}$ is the parameter matrix that maps each patch into a $D$-dimensional embedding.

- *Frequency Transformer.* We then apply a frequency transformer that operates along the frequency axis of $\mathbf{S}_i$, to model intra-spectral window cross-frequency band dependencies.

- *Gated Patchwise Aggregation.* In many EEG scenarios, large portions of the frequency spectrum can be irrelevant. For instance, tasks related to sleep primarily focus on frequency bands up to approximately 32 Hz (Chen et al., 2023). Also, the frequencies of interest vary across conditions and tasks. To emphasize important frequency patches and suppress the rest, we adopt a gated aggregation mechanism to obtain a embedding for each $S_i$: $\mathbf{E}_i^F = \text{Concat}\left[\sigma\left(\mathbf{W_{g1}}\mathbf{e_{(i,p)}}\right)\mathbf{W_{g2}}\mathbf{e_{(i,p)}}\right]$ where $\mathbf{W_{g1}}, \mathbf{W_{g2}}$ are trainable parameters and $\sigma(\cdot)$ is the element-wise sigmoid function.

**Temporal Encoder and Temporal Transformer.** To capture temporal dynamics from raw EEG patches $\{x_i\}_{i=1}^N$, each patch is projected linearly, followed by GELU activation and group normalization, producing temporal embeddings $\{\mathbf{E}_i^T\}_{i=1}^N$. Each aggregated frequency embedding $\mathbf{E}_i^F$ is then concatenated with its corresponding temporal embedding $\mathbf{E}_i^T$, and the resulting sequence is processed by a temporal Transformer. This module integrates time and frequency features across $N$ EEG patches, enabling the modeling of long-range dependencies. Finally, the outputs $\mathbf{Z}_i$ are quantized into discrete tokens using a learnable vocabulary $\mathcal{V}^k$. Notably, we omit positional encoding because EEG signals are inherently non-stationary and often exhibit chaotic dynamics; our objective is to capture distinctive features without enforcing positional constraints (see Appendix C.6).

**VQ Tokenizer Vocabulary.** Our vocabulary is based on the discrete codebook of Vector-Quantized Variational Autoencoders (VQ-VAE). We perform vector quantization to fused embedding $\mathbf{Z}_i$ that enables the vocabulary to capture time–frequency motifs as discrete tokens, supporting timestamp-level retrieval and improving EEG interpretability. Formally, given $\mathbf{Z} = \{\mathbf{z}_i\}_{i=1}^N$, each $\mathbf{z}_i$ is mapped to the closest code in the codebook $\mathcal{V} = \{\mathbf{v}_1, \ldots, \mathbf{v}_K\}$ by nearest-neighbor search.

$$q(\mathbf{z}_i) = \arg \min_{\mathbf{v}_k \in \mathcal{V}} \|\mathbf{z}_i - \mathbf{v}_k\|_2^2.$$

where $K$ denotes the number of latent vectors in the codebook and defines a $K$-way discrete categorical distribution. Each patch $z_i$ is mapped to its nearest code entry $v_i$. As a result, given a single-channel EEG $\mathbf{X}^c$, TFM-Tokenizer generates a sequence of $N$ tokens $\{v_i\}_{i=1}^N$.

**Frequency Masking Prediction for Tokenizer Learning**

We employ a joint frequency–temporal masking strategy for TFM-Tokenizer training. The spectrogram $\mathbf{S}$ is partitioned along the frequency axis into $N_F = \lfloor F/\delta_f \rfloor$ groups of size $\delta_f$, and random frequency-band masks $M_F$ and temporal masks $M_T$ are applied to obtain the masked input $\mathbf{S}^M$. Following (Jiang et al., 2024b), we further adopt symmetric masking for data augmentation and training stability. The overall objective combines masked reconstruction and vocabulary loss:

$$\mathcal{L}_{\text{token}} = \sum_{(f,t)} \|\mathbf{S}(f,t) - \hat{\mathbf{S}}(f,t)\|_2^2 + \alpha \sum_i \|\text{sg}[E_i] - v_i\|_2^2 + \beta \sum_i \|E_i - \text{sg}[v_i]\|_2^2$$

where $\hat{\mathbf{S}}$ is the reconstruction, $\text{sg}[\cdot]$ is the stop-gradient operator, and $\alpha, \beta$ are hyperparameters. We also apply exponential moving average updates for stable codebook training.

### 3.3 DOWNSTREAM TRANSFORMER TRAINING

We employ a lightweight transformer model to aggregate tokenized representations across channels, learn cross-channel dependencies and perform downstream tasks. It consists of a token-embedding lookup table (initialized from the VQ codebook) followed by linear attention transformer layers. Given a multi-channel recording $\mathbf{X} \in \mathbb{R}^{C \times T}$, the pretrained TFM-Tokenizer produces token sequences $\left\{\{v_i^c\}_{i=1}^N\right\}_{c=1}^C$ for each channel $c$ independently. We flatten the token embeddings across channels and incorporate channel and position embeddings. An addtional class token is prepended (Devlin, 2018), and the sequence is processed by transformer layers.

In order to pretrain the model and enable the model to learn intra and cross-channel dependencies of tokens, we adopt a strategy akin to masked language modeling. We first randomly mask tokens across multiple channels and time steps and then train the model to predict these masked tokens via a cross-entropy loss. Along with representation learning, this approach enhances robustness to missing or corrupted data, common in real-world EEG systems where channels or time segments may be dropped or noisy. Finally, the transformer model is finetuned for downstream tasks.

## 4 EXPERIMENTS AND RESULTS

### 4.1 EXPERIMENT SETUP

**Datasets:** We evaluated our method on four EEG datasets. **(1) TUEV** (Harati et al., 2015): A subset of the TUH EEG Corpus (Obeid & Picone, 2016), containing clinical EEG recordings annotated for six event types: spike and sharp wave (SPSW), generalized periodic epileptiform discharges (GPED), periodic lateralized epileptiform discharges (PLED), eye movement (EYEM), artifact (ARTF), and background (BCKG). **(2) TUAB** (Lopez et al., 2015): Also from Temple University Hospital, labeled for normal and abnormal EEG activity. **(3) CHB-MIT** (Shoeb, 2009): A widely used benchmark for epilepsy seizure detection, comprising EEG recordings from 23 pediatric subjects with intractable seizures. **(4) IIIC Seizure** (Jing et al., 2023; Ge et al., 2021): Designed for detecting six ictal–interictal–injury continuum (IIIC) patterns, including others (OTH), electrographic seizures (ESZ), lateralized periodic discharges (LPD), generalized periodic discharges (GPD), lateralized rhythmic delta activity (LRDA), and generalized rhythmic delta activity (GRDA). - *Scalability Validation.* In this paper, we provided a scalability experiment to evalute the usability of our tokenizer across different EEG devices. Since our tokenizer is trained in a single-channel setting, it can naturally be applied to recordings from non-standard devices. Therefore, we evaluated on the **Ear-EEG Sleep Monitoring (EESM23)** (Bjarke Mikkelsen et al., 2025; Tabar et al., 2024) dataset, which contains ear-EEG sleep recordings from 10 subjects. Detailed dataset statistics, splits, and preprocessing procedures are provided in Appendix B.1, B.2, and B.3.

**Baselines:** We evaluated our approach against the baselines from Yang et al. (2024) and recent state-of-the-art methods, including BIOT, LaBraM, NeuroLM, and EEGPT. We adopted the best results reported in BIOT, except for the IIIC Seizure dataset, where we re-evaluated the methods due to a sample size mismatch. Experiments were conducted under two settings: (1) Single-dataset setting: pretraining and finetuning on the same single dataset, and (2) Multiple dataset setting: pretraining on four EEG datasets. For BIOT, we reproduced their unsupervised pretraining and finetuning pipeline in the single-dataset setting (denoted BIOT$^\star$) to enable a fair comparison, as their vanilla BIOT variant does not include pretraining. Similarly, we reproduced LaBraM by training its neural tokenizer, performing masked EEG modeling, and finetuning within the same dataset (LaBraM$^\star$). Since our focus is on EEG tokenization rather than full foundation modeling, we reproduced LaBraM under the multiple dataset setting using the previously mentioned four EEG datasets (denoted LaBraM$^\dagger$). This was necessary to ensure a fair comparison because the original LaBraM used a substantially larger pretraining corpus. Additional experiment details are provided in Appendix B.4 and B.5.

### 4.2 HOW DOES TFM-TOKENIZER COMPARE TO EXISTING BASELINES?

Table 1 reports results on TUEV (event classification) and TUAB (abnormal detection), while Table 2 summarizes performance on IIIC-Seizure (seizure type classification) and CHB-MIT (seizure detection). Our TFM-Tokenizer paired with a downstream transformer outperforms the baselines in both experiment settings. On the challenging six-class event-type classification task in TUEV, it achieves a 5% gain in Cohen's Kappa in the single-dataset setting and a notable $\sim 11\%$ improvement ($0.5588 \rightarrow 0.6189$) in the multi-dataset setting over the next best baseline. On IIIC-Seizure, which is another six-class classification task, TFM-Tokenizer improves Cohen's Kappa by 36% over the LaBraM ($0.3658 \rightarrow 0.4979$) and 3% improvement over CBraMod ($0.4792 \rightarrow 0.4979$) in multiple dataset settings, demonstrating the strong capability of our tokenizer in modeling class-discriminative features for complex clinical EEG tasks. Additionally, it is worth noting that TFM-Tokenizer achieves better performance with fewer parameters, being 3 times smaller than LaBraM and 1.5 times smaller than BIOT. The ability to achieve best performance with low model size can be attributed to our tokenization approach, which compresses the EEG into a token sequence, thereby reducing data complexity. Notably, the TFM-Tokenizer is paired with a lightweight transformer comprising only $\sim$0.7M parameters.

### 4.3 CAN TFM-TOKENIZER IMPROVE EXISTING FOUNDATION MODELS?

To evaluate the generalizability of TFM-Tokenizer, we integrated it into two representative EEG foundation models, BIOT and LaBraM, under both single- and multi-dataset settings. For BIOT, we replaced raw EEG inputs with token embeddings while following the original training protocol. For

Table 1: Performance comparison on TUEV and TUAB datasets.

| Models | Model Size | TUEV (event type classification) | | | TUAB (abnormal detection) | | |
|---|---|---|---|---|---|---|---|
| | | Balanced Acc. | Cohen's Kappa | Weighted F1 | Balanced Acc. | AUC-PR | AUROC |
| **Single Dataset Setting** | | | | | | | |
| SPaRCNet (Jing et al., 2023) | 0.79M | $0.4161 \pm 0.0262$ | $0.4233 \pm 0.0181$ | $0.7024 \pm 0.0104$ | $0.7896 \pm 0.0018$ | $0.8414 \pm 0.0018$ | $0.8676 \pm 0.0012$ |
| ContraWR (Yang et al., 2023) | 1.6M | $0.4384 \pm 0.0349$ | $0.3912 \pm 0.0237$ | $0.6893 \pm 0.0136$ | $0.7746 \pm 0.0041$ | $0.8421 \pm 0.0104$ | $0.8456 \pm 0.0074$ |
| CNN-Transformer (Peh et al., 2022) | 3.2M | $0.4087 \pm 0.0161$ | $0.3815 \pm 0.0134$ | $0.6854 \pm 0.0293$ | $0.7777 \pm 0.0022$ | $0.8433 \pm 0.0039$ | $0.8461 \pm 0.0013$ |
| FFCL (Li et al., 2022) | 2.4M | $0.3979 \pm 0.0104$ | $0.3732 \pm 0.0188$ | $0.6783 \pm 0.0120$ | $0.7848 \pm 0.0038$ | $0.8448 \pm 0.0065$ | $0.8569 \pm 0.0051$ |
| ST-Transformer (Song et al., 2021) | 3.5M | $0.3984 \pm 0.0228$ | $0.3765 \pm 0.0306$ | $0.6823 \pm 0.0190$ | $\underline{0.7966} \pm 0.0023$ | $0.8521 \pm 0.0026$ | $0.8707 \pm 0.0019$ |
| Vanilla BIOT (Yang et al., 2024) | 3.2M | $0.4682 \pm 0.0125$ | $0.4482 \pm 0.0285$ | $0.7085 \pm 0.0184$ | $0.7925 \pm 0.0035$ | $0.8707 \pm 0.0087$ | $0.8691 \pm 0.0033$ |
| BIOT* (Yang et al., 2024) | 3.2M | $0.4679 \pm 0.0354$ | $0.4890 \pm 0.0407$ | $0.7352 \pm 0.0236$ | $0.7955 \pm 0.0047$ | $\underline{0.8819} \pm 0.0046$ | $\underline{0.8834} \pm 0.0041$ |
| LaBraM-Base* (Jiang et al., 2024b) | 5.8M | $\underline{0.4682} \pm 0.0856$ | $\underline{0.5067} \pm 0.0413$ | $\underline{0.7466} \pm 0.0202$ | $0.7720 \pm 0.0046$ | $0.8498 \pm 0.0036$ | $0.8534 \pm 0.0027$ |
| **TFM-Tokenizer (Ours)** | 1.9M | $\mathbf{0.4943} \pm 0.0516$ | $\mathbf{0.5337} \pm 0.0306$ | $\mathbf{0.7570} \pm 0.0163$ | $\mathbf{0.8152} \pm 0.0014$ | $\mathbf{0.8946} \pm 0.0008$ | $\mathbf{0.8897} \pm 0.0008$ |
| **With Multiple Dataset Pretraining** | | | | | | | |
| BIOT (Yang et al., 2024) | 3.2M | $0.5281 \pm 0.0225$ | $0.5273 \pm 0.0249$ | $0.7492 \pm 0.0082$ | $0.7959 \pm 0.0057$ | $\underline{0.8792} \pm 0.0023$ | $\underline{0.8815} \pm 0.0043$ |
| EEGPT (Wang et al., 2024a) | 4.7M | $0.5670 \pm 0.0066$ | $0.5085 \pm 0.0173$ | $0.7535 \pm 0.0097$ | $\underline{0.7959} \pm 0.0021$ | - | $0.8716 \pm 0.0041$ |
| NeuroLM-B (Jiang et al., 2024a) | 254M | $0.4560 \pm 0.0048$ | $0.4285 \pm 0.0048$ | $0.7153 \pm 0.0028$ | $0.7826 \pm 0.0065$ | $0.6975 \pm 0.0081$ | $0.7816 \pm 0.0079$ |
| LaBraM-Base† (Jiang et al., 2024b) | 5.8M | $0.5550 \pm 0.0403$ | $0.5175 \pm 0.0339$ | $0.7450 \pm 0.0194$ | $0.7735 \pm 0.0030$ | $0.8531 \pm 0.0028$ | $0.8557 \pm 0.0027$ |
| CBraMod† (Wang et al., 2024d) | 4M | $\underline{0.5696} \pm 0.0221$ | $\underline{0.5588} \pm 0.0273$ | $\underline{0.7702} \pm 0.0137$ | $0.5000 \pm 0.0000$ | $0.4938 \pm 0.0443$ | $0.5281 \pm 0.0409$ |
| **TFM-Tokenizer (Ours)** † | 1.9M | $\mathbf{0.5974} \pm 0.0079$ | $\mathbf{0.6189} \pm 0.0302$ | $\mathbf{0.8010} \pm 0.0161$ | $\mathbf{0.8032} \pm 0.0035$ | $\mathbf{0.8886} \pm 0.0032$ | $\mathbf{0.8870} \pm 0.0022$ |

Table 2: Performance comparison on IIIC Seizure and CHB-MIT datasets.

| Models | Model Size | IIIC Seizure (seizure type classification) | | | CHB-MIT (seizure detection) | | |
|---|---|---|---|---|---|---|---|
| | | Balanced Acc. | Cohen's Kappa | Weighted F1 | Balanced Acc. | AUC-PR | AUROC |
| **Single Dataset Setting** | | | | | | | |
| SPaRCNet (Jing et al., 2023) | 0.79M | $0.5011 \pm 0.0286$ | $0.4115 \pm 0.0297$ | $0.4996 \pm 0.0262$ | $0.5876 \pm 0.0191$ | $0.1247 \pm 0.0119$ | $0.8143 \pm 0.0148$ |
| ContraWR (Yang et al., 2023) | 1.6M | $0.5421 \pm 0.0123$ | $0.4549 \pm 0.0166$ | $0.5387 \pm 0.0138$ | $0.6344 \pm 0.0002$ | $0.2264 \pm 0.0174$ | $0.8097 \pm 0.0114$ |
| CNN-Transformer (Peh et al., 2022) | 3.2M | $0.5395 \pm 0.0144$ | $0.4500 \pm 0.0165$ | $0.5413 \pm 0.0176$ | $0.6389 \pm 0.0067$ | $0.2479 \pm 0.0227$ | $\underline{0.8662} \pm 0.0082$ |
| FFCL (Li et al., 2022) | 2.4M | $0.5309 \pm 0.0217$ | $0.4412 \pm 0.0253$ | $0.5315 \pm 0.0277$ | $0.6262 \pm 0.0104$ | $0.2049 \pm 0.0346$ | $0.8271 \pm 0.0051$ |
| ST-Transformer (Song et al., 2021) | 3.5M | $0.5093 \pm 0.0122$ | $0.4217 \pm 0.0151$ | $0.5217 \pm 0.0110$ | $0.5915 \pm 0.0195$ | $0.1422 \pm 0.0094$ | $0.8237 \pm 0.0491$ |
| Vanilla BIOT (Yang et al., 2024) | 3.2M | $\underline{0.5762} \pm 0.0034$ | $\underline{0.4932} \pm 0.0046$ | $\underline{0.5773} \pm 0.0031$ | $\underline{0.6640} \pm 0.0037$ | $0.2573 \pm 0.0088$ | $0.8646 \pm 0.0030$ |
| BIOT* (Yang et al., 2024) | 3.2M | $0.4458 \pm 0.0183$ | $0.3418 \pm 0.0228$ | $0.4511 \pm 0.0207$ | $0.6582 \pm 0.0896$ | $\underline{0.3127} \pm 0.0890$ | $0.8456 \pm 0.0333$ |
| LaBraM-Base* (Jiang et al., 2024b) | 5.8M | $0.4736 \pm 0.0101$ | $0.3716 \pm 0.0128$ | $0.4765 \pm 0.0097$ | $0.5035 \pm 0.0078$ | $0.0959 \pm 0.0742$ | $0.6624 \pm 0.1050$ |
| **TFM-Tokenizer (Ours)** | 1.9M | $\mathbf{0.5775} \pm 0.0042$ | $\mathbf{0.4985} \pm 0.0039$ | $\mathbf{0.5847} \pm 0.0050$ | $\mathbf{0.6750} \pm 0.0392$ | $\mathbf{0.3379} \pm 0.0515$ | $\mathbf{0.8839} \pm 0.0173$ |
| **With Multiple Dataset Pretraining** | | | | | | | |
| BIOT (Yang et al., 2024) | 3.2M | $0.4414 \pm 0.0035$ | $0.3362 \pm 0.0040$ | $0.4483 \pm 0.0033$ | $\mathbf{0.7068} \pm 0.0457$ | $0.3277 \pm 0.0460$ | $0.8761 \pm 0.0284$ |
| EEGPT (Wang et al., 2024a) | 4.7M | $0.4545 \pm 0.0193$ | $0.3502 \pm 0.0255$ | $0.4559 \pm 0.0311$ | $0.6644 \pm 0.0227$ | $0.3373 \pm 0.0264$ | $0.8185 \pm 0.0252$ |
| LaBraM-Base† (Jiang et al., 2024b) | 5.8M | $0.4736 \pm 0.0037$ | $0.3658 \pm 0.0033$ | $0.4708 \pm 0.0015$ | $0.5260 \pm 0.0369$ | $0.2138 \pm 0.0523$ | $0.7750 \pm 0.0540$ |
| CBraMod† (Wang et al., 2024d) | 4M | $\underline{0.5566} \pm 0.0126$ | $\underline{0.4792} \pm 0.0167$ | $\underline{0.5743} \pm 0.0138$ | $0.6646 \pm 0.0598$ | $\underline{0.3469} \pm 0.0281$ | $\mathbf{0.9071} \pm 0.0199$ |
| **TFM-Tokenizer (Ours)** † | 1.9M | $\mathbf{0.5747} \pm 0.0022$ | $\mathbf{0.4979} \pm 0.0038$ | $\mathbf{0.5797} \pm 0.0017$ | $\underline{0.6471} \pm 0.0145$ | $\mathbf{0.3554} \pm 0.0264$ | $\underline{0.8818} \pm 0.0117$ |

1. The best and second-best results for each dataset setting are **bolded** and underlined, respectively. 2. The number of parameters for LaBraM is only considering their classifier model. The size of their neural tokenizer was 8.6M. 3. ⋆ indicates reproduced in a single dataset setting and † indicates pretraining on 4 EEG datasets.

LaBraM, we substituted its neural tokenizer with ours during masked EEG modeling. As shown in Figure 3, our method consistently improves performance on TUEV, IIIC, and CHB-MIT, achieving gains of at least 3% in most cases. LaBraM notably underperforms on CHB-MIT in the single-dataset setting, yet integrating our tokenizer yields a 147% improvement in AUC-PR, demonstrating its effectiveness in capturing class-discriminative features in data-scarce scenarios. These results highlight the broad applicability of TFM-Tokenizer across architectures and its capacity to enhance diverse EEG foundation models.

## 4.4 DOES TFM-TOKENIZER SCALE TO OTHER BRAIN-SIGNAL TYPES / DEVICES?

In order to assess the scalability of TFM-Tokenizer beyond the modalities and tasks seen during pretraining, we evaluate its performance on the EESM23 ear-EEG dataset (Bjarke Mikkelsen et al., 2025) for sleep staging, a task, brain signal modality, acquisition system, number of channels and channel configuration entirely distinct from those in the pretraining set. Specifically, we only finetune pretrained

Table 3: Scalability experiments results on EESM23.

| Models | Ear-EEG (Sleep Staging) | | |
|---|---|---|---|
| | Balanced Acc. | Cohen's Kappa | Weighted F1 |
| BIOT | $0.3858 \pm 0.0085$ | $0.3406 \pm 0.0096$ | $0.4888 \pm 0.0124$ |
| BIOT-TFM | $0.3952 \pm 0.0170 \uparrow$ | $0.3603 \pm 0.0252 \uparrow$ | $0.5033 \pm 0.0165 \uparrow$ |
| LaBraM-Base | $0.3890 \pm 0.0182$ | $0.3322 \pm 0.0232$ | $0.4827 \pm 0.0157$ |
| LaBraM-TFM | $0.4004 \pm 0.0086 \uparrow$ | $0.3475 \pm 0.0128 \uparrow$ | $0.4864 \pm 0.0118 \uparrow$ |
| **TFM-Tokenizer** | $\mathbf{0.4148} \pm 0.0209$ | $\mathbf{0.3883} \pm 0.0233$ | $\mathbf{0.5174} \pm 0.0141$ |

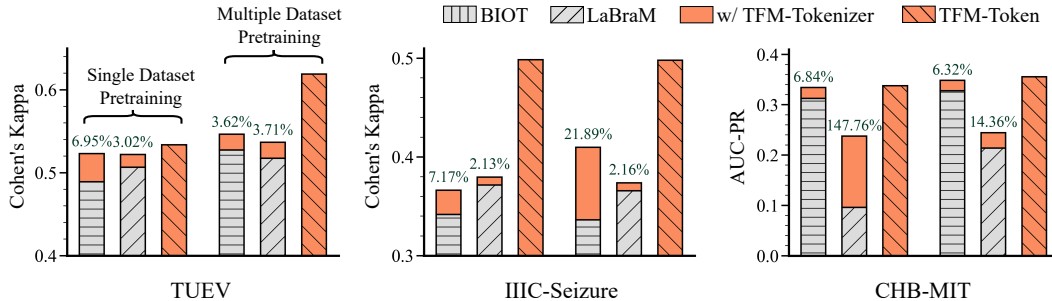

Figure 3: Performance comparison of existing foundation models with and without integration of `TFM-Tokenizer` on the TUEV, IIIC, and CHB-MIT datasets. For each dataset, the first three bars show single-dataset pretraining and the latter three show multi-dataset pretraining. Percentage values above each bar indicate the relative performance gain achieved by incorporating TFM-Tokenizer.

models (our method, BIOT, and LaBraM) on the EESM23 dataset using only ∼8K labeled training samples. EEGPT was not scalable in this setting due to its reliance on a fixed EEG channel layout for spatial embeddings (Wang et al., 2024a). As shown in Table 3, TFM-Tokenizer demonstrates strong generalization, outperforming both baselines ($p = 0.02$) in this out-of-domain setting.

## 4.5 HOW IMPORTANT ARE FREQUENCY AND TEMPORAL MODELING FOR EEG TOKENIZATION?

To evaluate the importance of joint frequency–temporal modeling, we conducted an ablation study with three tokenization variants: (1) TFM-Tokenizer-R, which uses only raw EEG patches to predict the masked spectrogram; (2) TFM-Tokenizer-S, which uses only the spectrogram as input; and (3) TFM-Tokenizer, which jointly models both domains. Masked modeling was applied for token learning in the latter two. On TUEV (Figure 4a), TFM-Tokenizer-S achieves higher Cohen's Kappa than TFM-Tokenizer-R, while TFM-Tokenizer-R yields better AUC-PR in abnormal detection (Appendix Figure 6). These results show that different EEG tasks rely on different feature domains, underscoring the need for joint modeling, where TFM-Tokenizer consistently outperforms both variants.

## 4.6 HOW EFFECTIVE ARE TFM-TOKENIZER TOKENS?

We evaluate the quality of EEG tokens learned by our tokenizer across four aspects: (1) class-specific distinctiveness, (2) token consistency, (3) frequency learning capability, and (4) token utilization (results in Appendix C.1). For this analysis, we compare all three TFM-Tokenizer variants with the neural tokenizer from LaBraM, using the test splits of TUEV and IIIC, which both contain multiple classes. To ensure fairness, all tokenizers employ a fixed vocabulary size of 8192. Results on TUEV are shown in Figure 4b–c, with additional results for other datasets provided in the Appendix.

**Class-Token uniqueness.** To assess whether tokenizers capture class-specific motifs, we define the *Class-Token Uniqueness Score* as $\frac{\text{\# Unique Tokens in Class}}{\text{\# Tokens Utilized by Class}} \times 100\%$. This metric quantifies how well a tokenizer assigns distinctive tokens to each class. Figure 4b shows the scores for TUEV, where a robust tokenizer should yield high distinctiveness across all classes through unsupervised pretraining. TFM-Tokenizer consistently achieves higher scores than its variants and LaBraM's neural tokenizer, indicating that it produces more compact and informative token representations and validating the benefit of joint frequency–temporal modeling in EEG analysis.

**Class-wise Token Consistency Analysis.** We conduct a retrieval-based EEG signal mining experiment to evaluate token consistency within the same class, using similar-class sample retrieval (see Figure 4c). Given a multi-channel EEG sample, we first obtain its discrete token representation. Using the Jaccard similarity score, we then retrieve the top $K$ most similar samples from the dataset and compute the precision score for correctly retrieving samples of the same class. For this study, we constructed a balanced subset from the IIIC and TUEV datasets and tested all four tokenization methods. Results show that all TFM-Tokenizer variants significantly outperform the neural tokenizer. Among all variants, our method yields the best retrieval performance, reflecting better token consistency. Notably, TFM-Tokenizer-S and TFM-Tokenizer achieve nearly 60% precision on the TUEV for $K = 1$. While the Jaccard similarity measure demonstrates initial feasibility, further

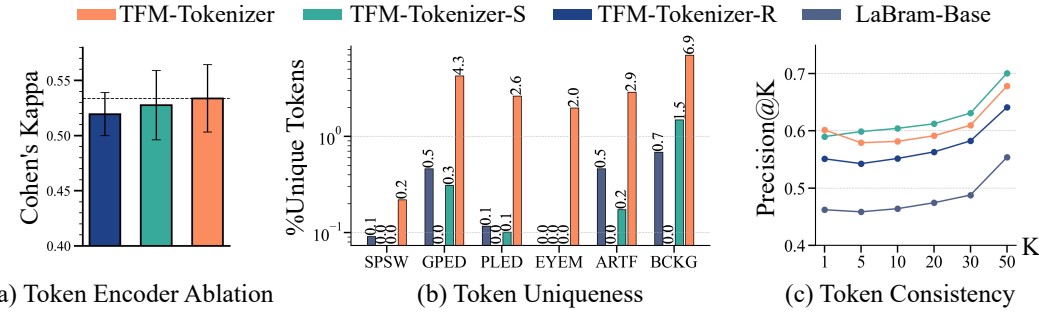

(a) Token Encoder Ablation      (b) Token Uniqueness      (c) Token Consistency

Figure 4: (a) Frequency and temporal token encoder ablation on TUEV. (b) Comparison of class-token uniqueness scores across all classes and (c) Class-wise token consistency analysis.

work is needed to identify optimal metrics. Nonetheless, the results suggest that EEG tokens can support the identification of similar pairs, with potential applications in contrastive learning.

### 4.7 DO THE LEARNED TOKENS CAPTURE MEANINGFUL EEG MOTIFS?

We perform a small-scale qualitative analysis to examine whether TFM-Tokenizer captures meaningful time–frequency motifs in EEG signals. Figure 5 shows some representative tokens learned by our method on the TUEV dataset. Each token represents a spectral window and its corresponding raw EEG patch (1s window with 0.5s overlap). For clarity, we highlight the most frequent tokens per class using distinct colors. Periodic Lateralized Epileptiform Discharges (PLEDs) are periodic patterns consisting of sharp waves or spikes followed by a slow wave, occurring every 1-2s (Pohlmann-Eden et al., 1996). Token 4035 consistently captures this characteristic waveform across different samples in the PLED class, despite variations in noise, amplitude, and minor temporal shifts. This confirms that our TFM-Tokenizer can capture class-specific

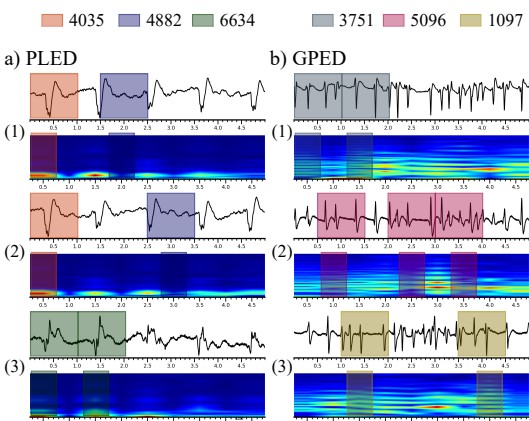

Figure 5: Overview of motifs captured by TFM-Tokenizer on TUEV: (a) three samples from the PLED class and (b) three samples from the GPED.

physiologically meaningful EEG motifs into discrete tokens. Similarly, tokens such as 5096 and 3751 in the GPED class highlight the benefit of joint time–frequency modeling, as they remain robust to minor temporal shifts and warping within a window due to emphasizing spectral patterns. However, we found limitations associated with using fixed windowing for tokenization, as large patterns or shifts may cause splits across windows, leading to separate token assignments and misinterpretation as distinct events.

## 5 CONCLUSION

In this paper, we presented `TFM-Tokenizer`, a model-agnostic tokenization framework that encodes *single-channel* EEG into discrete tokens by capturing time–frequency motifs. Our study demonstrated three key benefits: (i) Accuracy: By accurately extracting single-channel features, our tokenizer enabled stronger representations and surpassed competitive baselines across four EEG benchmarks. (ii) Generalization: As a plug-and-play component, our method consistently boosted the performance of existing foundation models, showing its broad applicability. (iii) Scalability: Because it operates at the single-channel level rather than depending on the strict 10–20 EEG system, our method readily extended to ear-EEG sleep staging tasks, validating its cross-device scalability. Furthermore, analyses confirmed the class distinctiveness, consistency, and interpretability of the learned tokens, providing deeper insights into EEG tokenization. We hope this work will inspire the development of more robust tokenization frameworks and advance scalable, generalizable EEG foundation models across diverse modalities, devices, and tasks.

ACKNOWLEDGMENT

This work was supported by NSF awards SCH-2205289, SCH-2014438, IIS-2034479, and JSPS KAKENHI Grant-in-Aid for Scientific Research Number JP24K20778. This project has been funded by the Jump ARCHES endowment through the Health Care Engineering Systems Center.

REPRODUCIBILITY STATEMENT

To support the reproducibility of our work, we provide our complete source code and pretrained model weights at https://github.com/Jathurshan0330/TFM-Tokenizer The repository includes scripts for data preprocessing, loading, and model training to reproduce our results presented in this paper. In the main text, Section 4.1 outlines our experimental setup, including descriptions of the dataset and baselines. Additional implementation details, such as dataset statistics, preprocessing steps, ear-EEG-specific processing, evaluation metrics, and baseline configurations, are provided in Appendix B.1, B.2, B.3, B.4, and B.5. The Appendix also includes extended experiments across multiple datasets, including frequency learning analysis (Appendix C.1), cross-dataset generalization studies (Appendix C.3), additional results on improving foundation models (Appendix C.4), and further ablation studies. We have made every effort to ensure that our work can be easily reproduced by the community.

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

APPENDIX

**Contents**

## A  PROBLEM FORMULATION

**EEG Data.** Let $\mathbf{X} \in \mathbb{R}^{C \times T}$ denote a multi-channel EEG recording with $C$ channels and $T$ time samples. Each channel $x^c \in \mathbb{R}^T$ is decomposed into (1) raw patches $\{x_i\}_{i=1}^N$ and (2) corresponding

time-frequency representation windows $\{\mathbf{S}_i\}_{i=1}^{N}$, where $N$ is the number of time windows. For simplicity, we omit the channel index and refer to $x$ as a single-channel EEG signal unless stated otherwise. To obtain the time-frequency representation, i.e., spectrogram, $\mathbf{S}$, we apply the short-time Fourier transform (STFT) to $x$ using a windowing function $w(.)$ of length $L$ and a hop size $H$.

**Short-Time Fourier Transform (STFT).** To obtain the time-frequency representation, i.e.g, spectrogram, $\mathbf{S}$, we apply a STFT to $x$ using a windowing function $w(.)$ of length $L$ and a hop size $H$:

$$\mathbf{S}(\omega, \tau) = \left| \sum_{l=0}^{L-1} x(\tau H + l) w(l) e^{\frac{-j2\pi \omega l}{L}} \right| \tag{1}$$

where $\omega$ indexes the discrete frequencies and $\tau$ indexes the time segments (i.e., time windows shifted by $H$). We retain only the magnitude $|.|$ to form $\mathbf{S} \in \mathbb{R}^{F \times N}$, where $F$ is the number of frequency bins and $N$ is the number of time windows.

**Problem Statement 1 (EEG Tokenization):** Given a single channel EEG $x$, we aim to learn a tokenization function $f_{\textbf{tokenizer}} : \mathbb{R}^T \to \mathcal{V}^{N \times D}$, that maps $x$ (or transformations) to a sequence of discrete tokens $\{v_i\}_{i=1}^{N}$, where each token is from a learnable EEG token vocabulary $\mathcal{V}$ of size $k$ and embedding size of $D$. These tokens should represent various time-frequency "*motifs*" derived from both $x_i$ and $\mathbf{S}_i$. Therefore, $\mathcal{V}$ is learnable from $\mathbf{S}$ and the temporal patches $\{x_i\}_{i=1}^{N}$. **Remark.** We here hold several expectations for the learned motif tokens. First, these tokens are expected to reduce redundancy, noise, and complexity, providing a compact, sparse, and informative representation of EEGs. Second, these motifs should capture key neurophysiological patterns from both temporal and frequency domains. Third, the tokens should generalize well across different EEG tasks.

**Problem Statement 2 (Multi-Channel EEG Classification):** Given EEGs $\mathbf{X}$ and a fixed, learned single-channel tokenizer $f_{\text{tokenizer}}$, we apply $f_{\text{tokenizer}}$ independently to each channel $c$ to obtain a tokenization representation $\left\{ \{v_i^c\}_{i=1}^{N} \right\}_{c=1}^{C}$. These tokens are aggregated and mapped to output labels by: $f_{\textbf{classifier}} : (\mathcal{V}^D)^{N \times C} \to \mathbf{Y}$ where $Y$ is the target labels (e.g., EEG events, seizure types). Notably, $f_{\text{classifier}}$ can be any downstream model, and its training is performed separately from the EEG tokenizer $f_{\text{tokenizer}}$.

# B    ADDITIONAL EXPERIMENT DETAILS

## B.1    DATASET STATISTICS AND SPLITS

Table 4: Evaluation Dataset Summary

| Dataset | # of Recordings | # of Samples | Duration (s) | Task |
|---------|-----------------|--------------|--------------|------|
| TUEV | $11,914$ | $112,491$ | 5 | EEG Event Classification |
| IIIC Seizure | $2,689$ | $135,096$ | 10 | Seizure Type Classification |
| CHB-MIT | 686 | $326,993$ | 10 | Seizure Detection |
| TUAB | $2,339$ | $409,455$ | 10 | Abnormal EEG Detection |
| EESM23 | 120 | $14,509$ | 30 | Ear-EEG based Sleep Staging |

This section provides detailed information on the datasets used in our experiments and their respective splits. Table 4 summarizes key statistics, including the number of recordings, the total number of samples after preprocessing, their duration, and the corresponding downstream tasks. For TUEV and TUAB, we utilized the official training and test splits provided by the dataset and further divided the training splits into $80\%$ training and $20\%$ validation sets. We performed a subject-wise split into $60\%$ training, $20\%$ validation, and $20\%$ test on the IIIC Seizure dataset. In the CHB-MIT dataset, we used 1-19 subjects for training, 20-21 for validation, and 22-23 for testing. For the out-of-distribution evaluation on the ear-EEG EESM23 (Bjarke Mikkelsen et al., 2025) dataset, we followed a subject-wise split, where subjects 1–6 were used for fine-tuning, 7–8 for validation, and 9–10 for testing.

## B.2 PREPROCESSING

We follow the preprocessing setup of BIOT (Yang et al., 2024). We adhere to the 16-channel bipolar montage from the international 10–20 system, as used in (Yang et al., 2024). All EEG recordings are resampled to 200 Hz. For TUEV and TUAB, we apply a bandpass filter (0.1–75 Hz) and a notch filter (50 Hz), following the preprocessing pipeline of LaBraM (Jiang et al., 2024b). We then segment the recordings according to the provided annotations and preprocessing guidelines. STFT computation of the signals is performed using PyTorch, with detailed parameters provided in Appendix B.6. For training, validation, and test splits, we follow the recommendations from (Yang et al., 2024). We adopt a window length of 1s with 0.5s overlap to segment EEG signals during training and inference, following prior work for consistency (Yang et al., 2024).

## B.3 EAR-EEG PREPROCESSING

We follow the preprocessing guidelines of Tabar et al. (2021) for the EESM-23 ear-EEG dataset, which includes four channels (RB, RT, LB, LT). A bandpass filter (0.1–100 Hz) and a 50Hz notch filter are applied. Each patients perform certain tasks before sleep. To isolate sleep segments, we crop each session from the onset of annotated sleep scoring, segment the signal into 30-second epochs, and discard corrupted segments.

## B.4 EVALUATION METRICS

For evaluation, we used balanced accuracy, Cohen's kappa coefficient, and weighted F1 for multi-class classification, and balanced accuracy, AUROC, and AUC-PR for binary classification. During finetuning, we employed binary cross-entropy loss for TUAB, cross-entropy loss for TUEV and IIIC, and focal loss for CHB-MIT due to class imbalance. All experiments were conducted using five different random seeds, and we report the mean and standard deviation.

## B.5 ADDITIONAL DETAILS ON BASELINES

All baselines were reproduced using their official open-source repositories. LaBraM's primary contribution lies in large-scale EEG pretraining using over 2,500 hours of data (Jiang et al., 2024b), whereas our focus is on developing an effective EEG tokenizer. To ensure a fair comparison, we reproduced LaBraM using its official repository under our dataset and experimental settings. For EEGPT, we report the published results for the 4.7M model on TUEV and TUAB (Wang et al., 2024a). Since results on CHB-MIT and IIIC-Seizure were not available, we used the official pretrained weights and fine-tuned the model on these tasks.

## B.6 STFT PARAMETERS

Table 5: STFT parameters

| Parameter | Value | Description |
|---|---|---|
| FFT size ($n_{\text{fft}}, L$) | 200 | Number of frequency bins (equal to resampling rate) |
| Hop length $H$ | 100 | Step size for sliding window (50% overlap) |
| Window type | Hann | A smoothing window function to reduce spectral leakage |
| Output representation | Magnitude | Only the absolute values of the STFT are retained |
| Centering | False | The STFT is computed without implicit zero-padding |
| One-sided output | True | Only the positive frequency components are kept |

To extract frequency-domain representations of the EEG, we utilized the STFT function from PyTorch. The recommendations of Yang et al. (2024) guided our parameter selection and empirical analysis of different configurations to optimize the trade-off between time-frequency resolution. The final parameters are as follows:

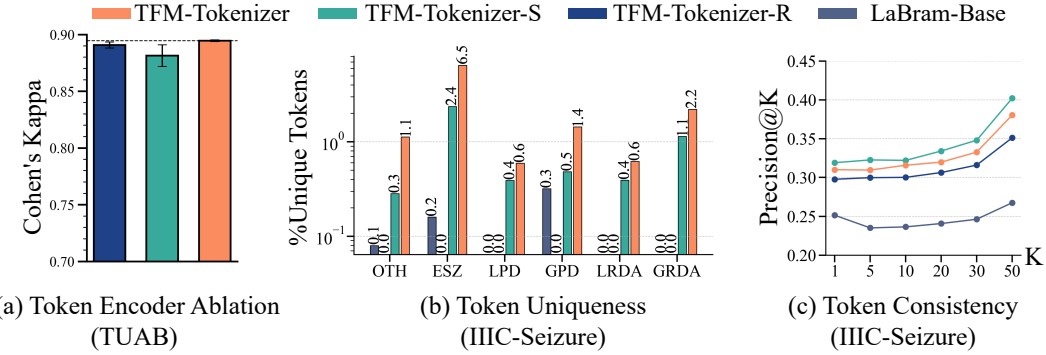

Figure 6: (a) Frequency and temporal token encoder ablation on TUAB. (b) & (c) presents Analysis of token quality across three `TFM-Tokenizer` variants and the neural tokenizer on IIIC. (b) Comparison of class-token uniqueness scores across all classes and (c) Class-wise token consistency analysis

Table 6: Token Utilization and class-token uniqueness comparison

| Tokenization Method | # Params | Utilization % | | Class-Token Uniqueness (GM) % | |
|---|---|---|---|---|---|
| | | **TUEV** | **IIIC** | **TUEV** | **IIIC** |
| Neural Tokenizer (LaBraM) | 8.6M | 21.13 | 15.25 | 0.034 | 0.000 |
| TFM-Tokenizer-R | 1.1M | 5.29 | 7.87 | 0.000 | 0.000 |
| TFM-Tokenizer-S | 1.1M | 13.93 | 11.04 | 0.004 | 0.619 |
| **TFM-Tokenizer** | 1.2M | 9.78 | 8.26 | 2.14 | 1.429 |

## C  EXTENDED EXPERIMENT RESULTS

### C.1  ADDITIONAL RESULTS ON TOKEN QUALITY ANALYSIS AND FREQUENCY LEARNING

In this section, we present more results on token quality analysis, specifically focusing on token utilization and frequency learning capability of our tokenizer. Additional token uniqueness and consistency experiments on IIIC dataset is presented in Figure 6b and c.

**Token utilization:** Token utilization (%) score was calculated as the percentage of unique tokens activated from the total available vocabulary size. Additionally, we computed the geometric mean (GM) of class-token uniqueness scores along with the utilization score, and the results are presented

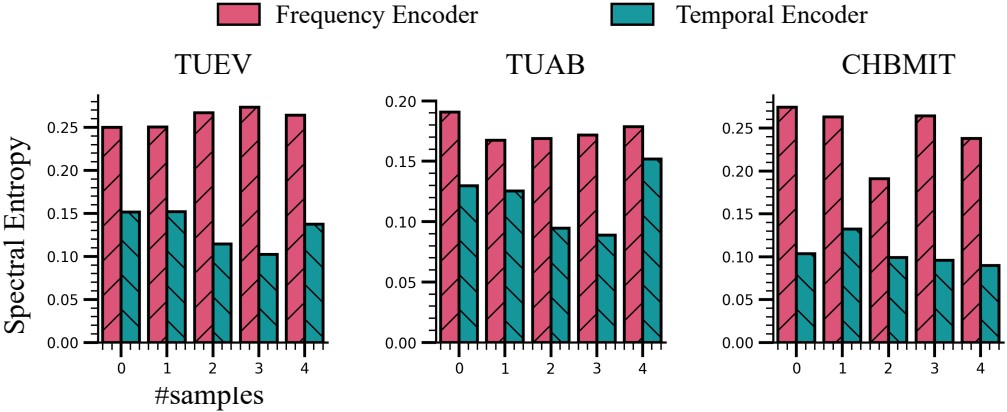

Figure 7: An analysis of how the proposed frequency and temporal-domain encoders capture frequency features, by using the spectral entropy of the learned token sequences from randomly selected samples. Higher values indicate that the tokens contain richer frequency information.

in Table 6. Our TFM-Tokenizer reduces token utilization by more than two-fold compared to the neural tokenizer on TUEV ($21.13\% \rightarrow 9.78\%$) and nearly two-fold on IIIC ($15.25\% \rightarrow 8.26\%$). It also significantly improves learning of class-unique tokens compared to the neural tokenizer ($0.034\% \rightarrow 2.14\%$ on TUEV, $0.0\% \rightarrow 1.429\%$ on IIIC).

**Evaluating the Frequency Learning of `TFM-Tokenizer` Tokens:** In this experiment, we compare the frequency and temporal-domain encoders of the TFM-Tokenizer to evaluate their ability to capture diverse frequency features in EEG signals. Specifically, we arrange all tokens in temporal order and perform a discrete Fourier transform on the token sequence. This process decomposes the tokens into frequencies, where each frequency reflects the degree of change between tokens at various scales. Larger changes indicate more diverse token representations. Then, we compute spectral entropy, defined as the normalized Shannon entropy of the amplitude values, to quantify how energy is distributed across the spectrum. Higher spectral entropy means that the model has learned a broader range of frequency features, capturing differences from both large-scale trends and fine details. Figure 7 shows that on the TUEV, TUAB, and CHBMIT datasets, the frequency encoder produces tokens with significantly higher spectral entropy than the temporal encoder. For example, on the TUEV dataset, the frequency encoder achieved an average spectral entropy of 0.26, while the temporal encoder reached only 0.14. This multi-scale sensitivity benefits downstream tasks such as classification, where learning detailed differences in EEG tokens can improve performance.

## C.2 Additional results on Frequency and Temporal Modeling for EEG Tokenization

Table 7: Ablation study on input representation to `TFM-Tokenizer`

| Models | TUEV (event type classification) | | | TUAB (abnormal detection) | | |
|---|---|---|---|---|---|---|
| | **Balanced Acc.** | **Cohen's Kappa** | **Weighted F1** | **Balanced Acc.** | **AUC-PR** | **AUROC** |
| TFM-Tokenizer-R | $\underline{0.4898} \pm 0.0105$ | $0.5194 \pm 0.0195$ | $0.7518 \pm 0.0095$ | $\underline{0.8033} \pm 0.0021$ | $\underline{0.8908} \pm 0.0027$ | $\underline{0.8849} \pm 0.0024$ |
| TFM-Tokenizer-S | $0.4708 \pm 0.0339$ | $\underline{0.5275} \pm 0.0314$ | $\underline{0.7538} \pm 0.0152$ | $0.7927 \pm 0.0044$ | $0.8814 \pm 0.0095$ | $0.8836 \pm 0.0052$ |
| **TFM-Tokenizer** | $\mathbf{0.4943} \pm \mathbf{0.0516}$ | $\mathbf{0.5337} \pm \mathbf{0.0306}$ | $\mathbf{0.7570} \pm \mathbf{0.0163}$ | $\mathbf{0.8152} \pm \mathbf{0.0014}$ | $\mathbf{0.8946} \pm \mathbf{0.0008}$ | $\mathbf{0.8897} \pm \mathbf{0.0008}$ |

1. The best results are **bolded**, while the second-best are underlined.

In Table 7 we provide detailed results of our ablation study discussed under Section 4.5.

## C.3 Token Generalization Assessment through Cross-Dataset Experiments

Table 8: Cross dataset generalizability experiments under single dataset settings

| Testing Dataset | Tokenizer Dataset | MTP Dataset | Performance Metrics | | |
|---|---|---|---|---|---|
| | | | **Balanced Acc.** | **Cohen's Kappa** | **Weighted F1** |
| TUEV | TUEV | TUEV | $0.4943 \pm 0.0516$ | $0.5337 \pm 0.0306$ | $0.7570 \pm 0.0163$ |
| | IIIC | TUEV | $0.4722 \pm 0.0578$ | $0.4990 \pm 0.0237$ | $0.7380 \pm 0.0137$ |
| | | IIIC | $0.4291 \pm 0.0235$ | $0.5195 \pm 0.0200$ | $0.7534 \pm 0.0100$ |
| | TUAB | TUEV | $0.4651 \pm 0.0449$ | $0.5925 \pm 0.0249$ | $0.7847 \pm 0.0136$ |
| | | TUAB | $0.5252 \pm 0.0431$ | $0.6187 \pm 0.0285$ | $0.8018 \pm 0.0138$ |
| | CHB-MIT | TUEV | $0.4979 \pm 0.0444$ | $0.5995 \pm 0.0225$ | $0.7885 \pm 0.0122$ |
| | | CHB-MIT | $\mathbf{0.5898} \pm 0.0192$ | $\mathbf{0.6591} \pm 0.0106$ | $\mathbf{0.8196} \pm 0.0045$ |

To evaluate the robustness of our tokenizer, we conducted cross-dataset experiments under two settings: (1) fixing the tokenizer and performing masked token prediction (MTP) & finetuning on a different target dataset and (2) fixing the tokenizer and MTP, followed by finetuning downstream transformer only on the target dataset. Results are presented in Table 8, which demonstrates strong generalizability, with our TFM-Tokenizer achieving the best performance on TUEV when pretrained on CHBMIT—outperforming the best-reported result in four dataset settings. These findings highlight the potential of our tokenizer as a foundation for a scalable, universal EEG tokenizer.

## C.4 Additional Results on TFM-Tokenizer Improving Existing Foundation Models

Table 9: Performance comparison of LaBraM and BIOT with and w/o our TFM-Tokenizer.

| Dataset | Exp. Setting | Method | Performance Metrics | | |
|---|---|---|---|---|---|
| | | | **Balanced Acc.** | **Cohen's Kappa** | **Weighted F1** |
| TUEV | Single | BIOT | $0.4679 \pm 0.0354$ | $0.4890 \pm 0.0407$ | $0.7352 \pm 0.0236$ |
| | | BIOT-TFM | $0.4228 \pm 0.0162$ | $0.5230 \pm 0.0226 \uparrow$ | $0.7490 \pm 0.0114 \uparrow$ |
| | | LaBraM | $0.4682 \pm 0.0856$ | $0.5067 \pm 0.0413$ | $0.7466 \pm 0.0202$ |
| | | LaBraM-TFM | $0.5147 \pm 0.0174 \uparrow$ | $0.5220 \pm 0.0153 \uparrow$ | $0.7533 \pm 0.0094 \uparrow$ |
| | Multiple | BIOT | $0.5281 \pm 0.0225$ | $0.5273 \pm 0.0249$ | $0.7492 \pm 0.0082$ |
| | | BIOT-TFM | $0.5530 \pm 0.0089 \uparrow$ | $0.5464 \pm 0.0137 \uparrow$ | $0.7625 \pm 0.0069 \uparrow$ |
| | | LaBraM | $0.5550 \pm 0.0403$ | $0.5175 \pm 0.0339$ | $0.7450 \pm 0.0194$ |
| | | LaBraM-TFM | $0.5541 \pm 0.0316$ | $0.5367 \pm 0.0281 \uparrow$ | $0.7567 \pm 0.0165 \uparrow$ |
| IIIC | Single | BIOT | $0.4458 \pm 0.0183$ | $0.3418 \pm 0.0228$ | $0.4511 \pm 0.0207$ |
| | | BIOT-TFM | $0.4633 \pm 0.0078 \uparrow$ | $0.3663 \pm 0.0103 \uparrow$ | $0.4689 \pm 0.0090 \uparrow$ |
| | | LaBraM | $0.4736 \pm 0.0101$ | $0.3716 \pm 0.0128$ | $0.4765 \pm 0.0097$ |
| | | LaBraM-TFM | $0.4814 \pm 0.0075 \uparrow$ | $0.3795 \pm 0.0091 \uparrow$ | $0.4841 \pm 0.0062 \uparrow$ |
| | Multiple | BIOT | $0.4414 \pm 0.0035$ | $0.3362 \pm 0.0040$ | $0.4483 \pm 0.0033$ |
| | | BIOT-TFM | $0.5050 \pm 0.0037 \uparrow$ | $0.4098 \pm 0.0052 \uparrow$ | $0.5139 \pm 0.0025 \uparrow$ |
| | | LaBraM | $0.4736 \pm 0.0037$ | $0.3658 \pm 0.0033$ | $0.4708 \pm 0.0015$ |
| | | LaBraM-TFM | $0.4782 \pm 0.0065 \uparrow$ | $0.3737 \pm 0.0076 \uparrow$ | $0.4790 \pm 0.0082 \uparrow$ |
| | | | **Balanced Acc.** | **AUC-PR** | **AUROC** |
| CHB-MIT | Single | BIOT | $0.6582 \pm 0.0896$ | $0.3127 \pm 0.0890$ | $0.8456 \pm 0.0333$ |
| | | BIOT-TFM | $0.5893 \pm 0.0197$ | $0.3341 \pm 0.0349 \uparrow$ | $0.8752 \pm 0.0123 \uparrow$ |
| | | LaBraM | $0.5035 \pm 0.0078$ | $0.0959 \pm 0.0742$ | $0.6624 \pm 0.1050$ |
| | | LaBraM-TFM | $0.5473 \pm 0.047 \uparrow$ | $0.2376 \pm 0.0461 \uparrow$ | $0.7863 \pm 0.0438 \uparrow$ |
| | Multiple | BIOT | $0.7068 \pm 0.0457$ | $0.3277 \pm 0.0460$ | $0.8761 \pm 0.0284$ |
| | | BIOT-TFM | $0.6197 \pm 0.0085$ | $0.3484 \pm 0.0078 \uparrow$ | $0.8726 \pm 0.0098$ |
| | | LaBraM | $0.5260 \pm 0.0369$ | $0.2138 \pm 0.0523$ | $0.7750 \pm 0.0540$ |
| | | LaBraM-TFM | $0.5579 \pm 0.0394 \uparrow$ | $0.2445 \pm 0.0351 \uparrow$ | $0.7887 \pm 0.0423 \uparrow$ |

Table 9 presents detailed results on integrating TFM-Tokenizer with BIOT and LaBraM. Across all metrics and settings, TFM-Tokenizer improves performance in 93% of cases, demonstrating its effectiveness in enhancing existing EEG foundation models.

## C.5 Effect of Masked Token Prediction in EEG Tokenization

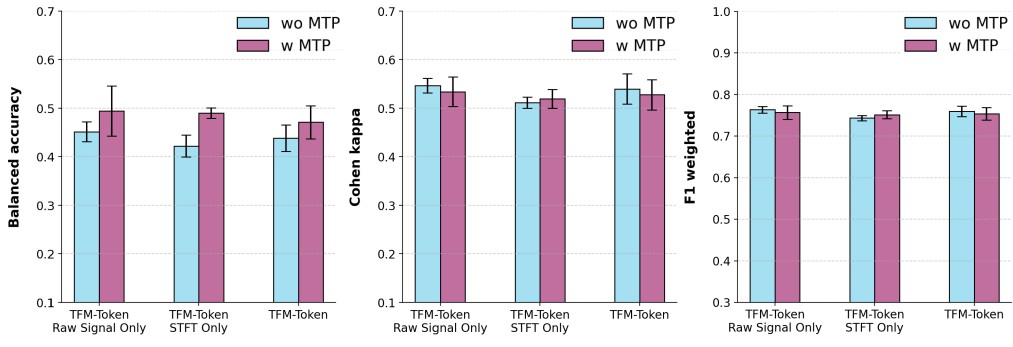

Figure 8: Masked Token Prediction Ablation

We conducted an ablation study on downstream transformer to assess the impact of masked token prediction pretraining in a fully discretized framework. Using a pretrained TFM-Tokenizer, we compared two approaches: (1) masked token prediction pretraining followed by fine-tuning and (2) direct fine-tuning without pretraining. This experiment was performed on the TUEV dataset across all three TFM-Tokenizer variants, with results summarized in Figure 8. While Cohen's Kappa and Weighted F1 showed no significant differences between the two approaches, masked token prediction pretraining significantly improved balanced accuracy across all TFM-Tokenizer variants. This suggests that pretraining enhances class-wise prediction consistency by capturing token dependencies and making downstream transformer more robust to missing channels or time segments, a common challenge in EEG analysis.

## C.6    REMOVING POSITION EMBEDDING IN `TFM-Tokenizer` IMPROVES TOKEN LEARNING

Table 10: TFM-Tokenizer Comparison with and w/o Position Embedding (PE) on TUEV Dataset

| Method | Utilization % | Uniqueness (GM) % | Balanced Acc. | Cohen's Kappa | Weighted F1 |
|---|---|---|---|---|---|
| `TFM-Tokenizer` + PE | 12.87 | 1.94 | $0.4765 \pm 0.038$ | $0.5119 \pm 0.022$ | $0.7457 \pm 0.012$ |
| `TFM-Tokenizer` w/o PE | 9.78 | 2.14 | $\mathbf{0.4943} \pm 0.052$ | $\mathbf{0.5337} \pm 0.031$ | $\mathbf{0.7570} \pm 0.016$ |

Through our empirical analysis, we found that the performance significantly improved when no position embedding was applied to the TFM-Tokenizer. EEG patterns are inherently chaotic and non-stationary, meaning similar motifs can occur at any position within the signal. An ideal tokenizer should be capable of capturing and representing such EEG motifs as distinct tokens without relying on positional information.

We conducted an ablation study comparing the TFM-Tokenizer's performance with and without position embeddings to critically analyze this phenomenon. The results of this analysis, presented in Table 10, clearly show that the TFM-Tokenizer without position embedding achieves significantly better performance, with an increase of $4\%$ in Cohen's Kappa ($0.5119 \rightarrow 0.5337$).

We further studied the quality of the learned tokens in terms of token utilization and class-uniqueness scores. Token utilization decreased ($12.87\% \rightarrow 9.78\%$) when position embeddings were removed, while the class-token uniqueness score increased ($1.94\% \rightarrow 2.14\%$). This suggests that the TFM-Tokenizer, when using positional encoding, learns different tokens for the same motifs depending on their location in the signal, leading to redundancy. Removing the position embedding allows the TFM-Tokenizer to learn more compact and meaningful tokens without introducing unnecessary data complexities. This improvement is further illustrated in the motifs captured by the TFM-Tokenizer's tokens in Figure 5 in Section 4.7.

## C.7    DOWNSTREAM MODEL ABLATION

We ablated the number of transformer layers in the downstream model on the TUEV dataset, with results presented in Table 11. Notably, even with significantly fewer parameters

Table 11: Ablation on number of transformer layers in the downstream model

| # Layers | Number of Params. | Performance Metrics | | |
|---|---|---|---|---|
| | | Balanced Acc. | Cohen's Kappa | Weighted F1 |
| 1 | 0.58M | $0.4486 \pm 0.0297$ | $0.5404 \pm 0.0168$ | $0.7603 \pm 0.0096$ |
| 2 | 0.63M | $0.4920 \pm 0.0595$ | $\mathbf{0.5758} \pm 0.0169$ | $\mathbf{0.7780} \pm 0.0089$ |
| 4 | 0.72M | $0.4943 \pm 0.0516$ | $0.5337 \pm 0.0306$ | $0.7570 \pm 0.0163$ |
| 6 | 0.82M | $\mathbf{0.5025} \pm 0.0592$ | $0.4996 \pm 0.0208$ | $0.7410 \pm 0.0104$ |
| 12 | 1.12M | $0.5016 \pm 0.0730$ | $0.5088 \pm 0.0272$ | $0.7456 \pm 0.0139$ |

(two layers), the model maintains competitive and, in some cases, better performance across key metrics. This highlights the potential for developing lightweight and efficient models for EEG analysis without substantial performance trade-offs.

## C.8 ABLATION ON TOKEN VOCABULARY SIZE

To evaluate the impact of token vocabulary size on performance and token learning, we conducted an ablation study by varying the vocabulary size from 256 to 8192 in powers of two. As shown in Figure 9, no monotonic trend was observed for Cohen's Kappa and Weighted F1 scores. However, balanced accuracy increased with larger vocabulary sizes. Further analysis of token utilization and class-token uniqueness scores is presented in Figure 10. Notably, Figure 10b shows that class-token uniqueness scores increase with vocabulary size, contributing to the improvement in balanced accuracy by enabling learning more unique class-specific tokens.

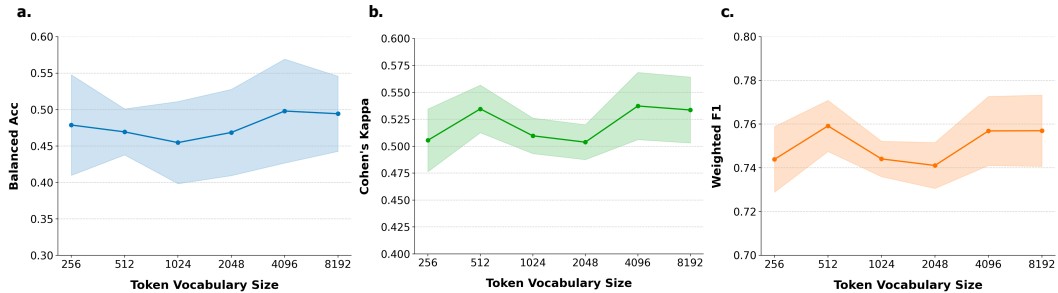

Figure 9: Token vocabulary size ablation with performance metrics

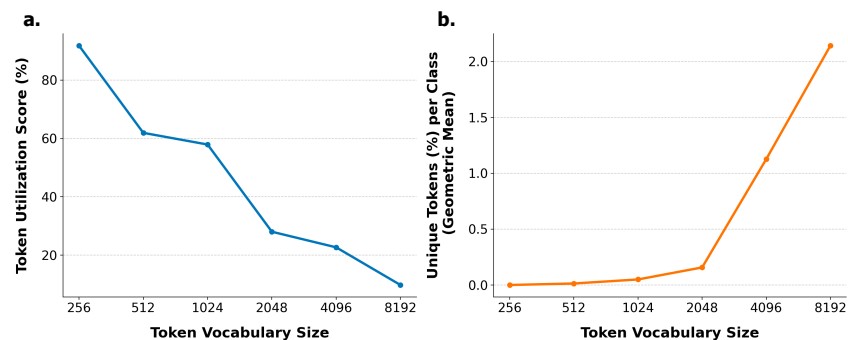

Figure 10: Token vocabulary size ablation with token utilization and uniqueness

## C.9 ABLATION ON MASKING

Table 12: Ablation on masking used for the pretraining of `TFM-Tokenizer` on TUEV Dataset

| Masking Strategy | Balanced Acc. | Cohen's Kappa | Weighted F1 |
|---|---|---|---|
| Random Masking | $0.4351 \pm 0.0462$ | $0.4772 \pm 0.0140$ | $0.7296 \pm 0.0076$ |
| Frequency Band Masking | $0.4673 \pm 0.0540$ | $0.5193 \pm 0.0243$ | $0.7536 \pm 0.0125$ |
| Frequency Band + Temporal Masking | $\mathbf{0.4946} \pm 0.0392$ | $0.5045 \pm 0.0221$ | $0.7462 \pm 0.0116$ |
| Frequency Band + Temporal Masking + Symmetric Masking | $0.4943 \pm 0.0516$ | $\mathbf{0.5337} \pm 0.0306$ | $\mathbf{0.7570} \pm 0.0163$ |

We conducted an ablation study on masking strategies during TFM-Tokenizer pretraining to assess their impact on performance. Results shown in Table 12 indicate that random masking on the spectrogram $S$ performs poorly compared to other strategies, underscoring the need for effective masking to capture frequency and temporal features from EEG. Frequency band masking

significantly improves performance over random masking, with an $8\%$ increase in Cohen's Kappa $(0.4772 \rightarrow 0.5193)$ and a $7\%$ increase in balanced accuracy $(0.4351 \rightarrow 0.4673)$, highlighting the importance of modeling frequency band dynamics. The addition of temporal masking further boosts balanced accuracy by $5\%$ $(0.4673 \rightarrow 0.4946)$, underscoring the importance of joint temporal-frequency modeling. However, temporal masking results in a decline in Cohen's Kappa and Weighted F1, which is then resolved by introducing symmetric masking, achieving the overall best performance.

## C.10 MASKING RATIO ABLATION

Table 13: Ablation on frequency band masking ratio used for the pretraining of `TFM-Tokenizer` on TUEV, IIIC Seizure and CHB-MIT Datasets.

| Dataset | Frequency Mask Ratio | Balanced Acc. | Cohen's Kappa | Weighted F1 |
|---|---|---|---|---|
| TUEV | 0.5 | **0.4946** $\pm$ 0.0392 | **0.5045** $\pm$ 0.0221 | **0.7462** $\pm$ 0.0116 |
| | 0.3 | 0.4306 $\pm$ 0.0187 | 0.5025 $\pm$ 0.0193 | 0.7432 $\pm$ 0.0090 |
| | 0.1 | 0.3859 $\pm$ 0.0580 | 0.4308 $\pm$ 0.0755 | 0.7057 $\pm$ 0.0376 |
| IIIC | 0.5 | **0.5315** $\pm$ 0.0102 | **0.4427** $\pm$ 0.0143 | **0.5369** $\pm$ 0.0114 |
| | 0.3 | 0.5148 $\pm$ 0.0158 | 0.4250 $\pm$ 0.0193 | 0.5222 $\pm$ 0.0167 |
| | 0.1 | 0.4381 $\pm$ 0.0032 | 0.3286 $\pm$ 0.0046 | 0.4420 $\pm$ 0.0047 |
| | | **Balanced Acc.** | **AUC-PR** | **AUROC** |
| CHB-MIT | 0.5 | **0.6809** $\pm$ 0.0380 | 0.3335 $\pm$ 0.0182 | **0.8859** $\pm$ 0.0137 |
| | 0.3 | 0.6313 $\pm$ 0.0599 | 0.3233 $\pm$ 0.0337 | 0.8708 $\pm$ 0.0187 |
| | 0.1 | 0.6530 $\pm$ 0.0486 | **0.3502** $\pm$ 0.0441 | 0.8742 $\pm$ 0.0116 |

We conducted an ablation study to examine how varying the frequency band masking ratio affects model performance and generalization across datasets. All experiments were performed under the single-channel setting, with the temporal masking ratio fixed at 0.5 without symmetric masking, and the results are summarized in Table 13. For the TUEV and IIIC Seizure datasets, a frequency mask ratio of 0.5 yielded the best overall performance. A similar trend was observed in the CHB-MIT dataset, except for Cohen's Kappa, which showed a slightly higher score at a masking ratio of 0.1. Considering these results along with the added benefit that a 0.5 masking ratio enables more effective use of symmetric masking as a data-augmentation strategy, we selected a frequency mask ratio of 0.5 for our final approach.

## C.11 WINDOW LENGTH ($L$) AND HOP SIZE ($H$) ABLATION

Table 14: Ablation on window length ($L$) and stride or hop size ($H$) used to segment raw signals and compute STFT for the pretraining of `TFM-Tokenizer` on TUEV Dataset.

| Window Length (s) | Hop Size (s) | Balanced Acc. | Cohen's Kappa | Weighted F1 |
|---|---|---|---|---|
| 0.5 | 0.25 | **0.5038** $\pm$ 0.0561 | **0.6059** $\pm$ 0.0170 | **0.7935** $\pm$ 0.0112 |
| 1.0 | 0.25 | 0.4796 $\pm$ 0.0598 | 0.5761 $\pm$ 0.0171 | 0.7780 $\pm$ 0.0098 |
| 1.0 | 0.5 | 0.4943 $\pm$ 0.0516 | 0.5337 $\pm$ 0.0306 | 0.7570 $\pm$ 0.0163 |
| 1.0 | 0.75 | 0.4068 $\pm$ 0.0182 | 0.4868 $\pm$ 0.0210 | 0.7327 $\pm$ 0.0085 |
| 2.0 | 0.5 | 0.1726 $\pm$ 0.0093 | 0.0168 $\pm$ 0.0137 | 0.5202 $\pm$ 0.0074 |
| 2.0 | 1.0 | 0.2123 $\pm$ 0.0143 | 0.1504 $\pm$ 0.0146 | 0.5748 $\pm$ 0.0087 |
| 2.0 | 1.5 | 0.3948 $\pm$ 0.0287 | 0.4042 $\pm$ 0.0282 | 0.6878 $\pm$ 0.0167 |

To investigate how window length and stride affect the tokenizer's ability to capture time–frequency motifs and performance, we conducted an ablation varying both parameters and adjusted the STFT configuration to preserve one-to-one alignment between time and frequency windows. The results, summarized in Table 14, indicate that smaller windows with greater overlap yield the strongest performance. This suggests that shorter segments allow the tokenizer to capture finer-grained motifs that may be lost when using larger windows. For consistency with baselines and prior work, however, we adopt a 1-second window length with a 0.5-second hop size in all reported experiments.

### C.12 Token Embedding Size Ablation

Table 15: Token embedding size ablation on TUEV Dataset.

| Embedding Dimension | Balanced Acc. | Cohen's Kappa | Weighted F1 |
|---|---|---|---|
| 32 | $0.4213 \pm 0.0529$ | $0.4974 \pm 0.0165$ | $0.7417 \pm 0.0081$ |
| 64 | $\mathbf{0.4943} \pm 0.0516$ | $\mathbf{0.5337} \pm 0.0306$ | $\mathbf{0.7570} \pm 0.0163$ |
| 128 | $0.3199 \pm 0.0193$ | $0.1909 \pm 0.0245$ | $0.5700 \pm 0.0276$ |
| 256 | $0.3864 \pm 0.0082$ | $0.3575 \pm 0.0157$ | $0.6682 \pm 0.0091$ |

Table 15 summarizes the results of the token embedding size ablation. Performance improves up to an embedding dimension of 64, after which it begins to decline. We do not observe a consistent trend as the embedding size increases, which may be attributed to training instability when using larger embedding dimensions.

## D  TFM-Tokenizer Implementation and Hyperparameter Tuning

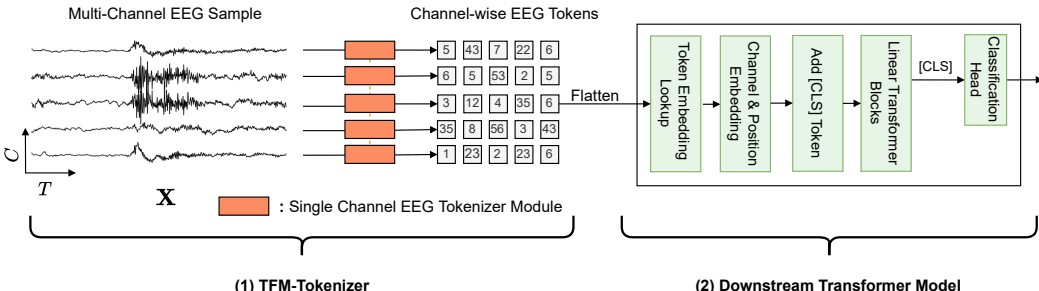

Figure 11: `TFM-Tokenizer` framework Overview

Figure 11 presents an overview of the framework during inference. This section provides additional details on the implementation and training of the framework.

### D.1 Hyperparameter Tuning of TFM-Tokenizer and Downstream Transformer

We employed a systematic approach to optimize the hyperparameters of both the TFM-Tokenizer and downstream transformer models using Ray Tune[1] with the Optuna[2] search algorithm. Our optimization process followed a three-phase strategy.

In the first phase, we optimized the TFM-Tokenizer architecture by tuning the depth and number of attention heads in the frequency transformer, temporal transformer, and transformer decoder modules to minimize the masked reconstruction loss $\mathcal{L}_{recon}$. This was followed by tuning the training optimizer's parameters, including learning rate and weight decay. The second phase focused on

---

[1] https://docs.ray.io/en/latest/tune/

[2] https://optuna.org/

the downstream transformer optimization for the classification task, where we first tuned its architectural parameters (depth and number of heads), followed by training the optimizer's parameters while keeping the tokenizer frozen. The third phase focused on tuning optimizer parameters for the masked token prediction pretraining of the downstream transformer.

To ensure a fair comparison with LaBraM's neural tokenizer, we maintained a vocabulary size of $8,192$ and an embedding dimension of $64$. For our ablation studies involving raw signal-only and STFT-only variants, we doubled the embedding dimensions of the temporal encoder and frequency patch encoder to match the codebook dimension while maintaining all other parameters same. Detailed hyperparameter configurations for both TFM-Tokenizer and downstream transformer are provided in Appendices D.2 and D.3, respectively.

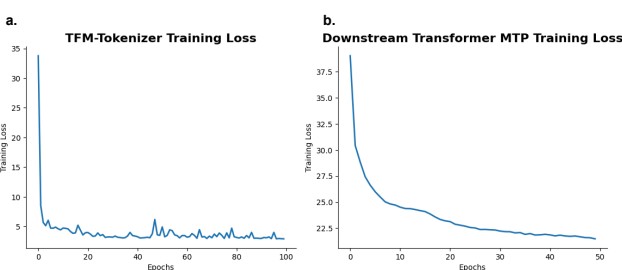

Figure 12: Training loss curves for (a) the TFM-Tokenizer learning and (b) the masked-token-prediction pretraining of the downstream transformer

In Figure 12a and b, we present the training loss curves for both the TFM-Tokenizer training stage and the masked-token-prediction pretraining of the downstream transformer, respectively. The curves demonstrate stable training behavior, even with a large codebook and a relatively small dataset. We kept the codebook size at 8192 to ensure a fair comparison with LaBraM's neural tokenizer.

## D.2 TFM-TOKENIZER HYPERPARAMETERS

Table 16: Hyperparameters for TFM-Tokenizer unsupervised pretraining on single-channel setting

| Hyperparameter | Values |
|---|---|
| Batch size | 256 |
| Optimizer | AdamW |
| Weight decay | 0.00001 |
| $\beta_1$ | 0.9 |
| $\beta_2$ | 0.99 |
| Learning rate scheduler | Cosine |
| Minimal Learning rate | 0.001 |
| Peak Learning rate | 0.005 |
| # of Warmup Epochs | 10 |
| # of Pretraining Epochs | 100 |

Table 17: Hyperparameters for TFM-Tokenizer

| Hyperparameter | | | Values |
|---|---|---|---|
| Temporal Encoder | Convolution layer 1 | Input Channels | 1 |
| | | Output Dimension | 64 |
| | | Kernel Size | 200 |
| | | Stride | 100 |
| | Convolution layer 2 | Output Dimension | 64 |
| | | Kernel Size | 1 |
| | | Stride | 1 |
| | Convolution layer 3 | Output Dimension | 32 |
| | | Kernel Size | 1 |
| | | Stride | 1 |
| Frequency Patch Encoder | Convolution layer 1 | Input Channels | 1 |
| | | Output Dimension | 64 |
| | | Kernel Size | 5 |
| | | Stride | 5 |
| | Convolution layer 2 | Output Dimension | 64 |
| | | Kernel Size | 1 |
| | | Stride | 1 |
| | Convolution layer 3 | Output Dimension | 64 |
| | | Kernel Size | 1 |
| | | Stride | 1 |
| Frequency Transformer | | Transformer Encoder Layers | 2 |
| | | Embedding Dimension | 64 |
| | | Number of Heads | 8 |
| Gated Patchwise Aggregation | | Output Dimension | 32 |
| | | Kernel Size | 5 |
| | | Stride | 5 |
| Temporal Transformer | | Transformer Encoder Layers | 2 |
| | | Embedding Dimension | 64 |
| | | Number of Heads | 8 |
| Token vocabulary (Codebook size) | | | 8192 |
| Transformer Decoder | | Transformer Encoder Layers | 8 |
| | | Embedding Dimension | 64 |
| | | Number of Heads | 8 |
| Linear Decoder | | | 100 |

## D.3 DOWNSTREAM TRANSFORMER ENCODER HYPERPARAMETERS

Table 18: Hyperparameters for downstream transformer, its masked token prediction pretraining and downstream finetuning

| Hyperparameter | Values |
|---|---|
| Transformer Encoder Layers | 4 |
| Embedding Dimension | 64 |
| Number of Heads | 8 |
| **Masked Token Prediction Pretraining** | |
| Batch size | 512 |
| Optimizer | AdamW |
| Weight decay | 0.00001 |
| $\beta_1$ | 0.9 |
| $\beta_2$ | 0.99 |
| Learning rate scheduler | Cosine |
| Minimal Learning rate | 0.001 |
| Peak Learning rate | 0.005 |
| # of Warmup Epochs | 5 |
| # of training Epochs | 50 |
| **Finetuning** | |
| Other parameters are the same as above except: | |
| $\beta_2$ | 0.999 |
| label smoothing (multi-class) | 0.1 |

# E   BIOT-TFM AND LaBraM-TFM IMPLEMENTATION DETAILS

Figure 13: Schematics of integrating the proposed TFM-Tokenizer with BIOT and LaBraM foundation models.

Figure 13 illustrates how the proposed TFM-Tokenizer is seamlessly integrated into existing EEG foundation models such as BIOT and LaBraM.

**BIOT-TFM.**   For BIOT, we replace the initial patch-based linear projection layers with our tokenizer, which converts multi-channel EEG signals into discrete token sequences. Similar to BIOT, we add channel embeddings and positional embeddings before feeding the tokens into the linear transformer layers. We follow BIOT's original training pipeline, which includes unsupervised pretraining, supervised pretraining, and downstream finetuning, allowing a direct comparison while isolating the effect of our tokenizer.

**LaBraM-TFM.**   LaBraM employs a neural tokenizer only during its masked EEG modeling (MEM) pretraining stage, where multi-channel EEG signals are patched, a subset of patches is masked, and the model is trained to predict the tokenizer-generated codebook indices for masked patches. In LaBraM-TFM, we simply replace LaBraM's neural tokenizer with our TFM-Tokenizer and conduct MEM pretraining as in the original workflow. After pretraining, the tokenizer is discarded and only the LaBraM transformer is finetuned for downstream tasks.

# F   MORE RELATED WORKS

**Frequency Representation Collapse.**   Frequency domain analysis is crucial in EEG and general time series analysis (Elvander & Jakobsson, 2020; Wu et al., 2021; 2023; Woo et al., 2022). In real-world signals, time-domain observations inherently mix multiple frequency components, and high-energy, low-frequency signals often dominate the spectrum (Huang Norden E Shen Zheng & H, 1998; Lai et al., 2018). As a result, these entangled frequency features makes it difficult for models to distinguish between them (Zhou et al., 2022; Piao et al., 2024). Recent studies have shown that these entangled signals can lead to a collapse in the learned frequency representations (Zhi-Qin John Xu et al., 2020; Piao et al., 2024). Models tend to overemphasize the dominant low-frequency

features while neglecting the high-frequency details. This issue can lead to a lack of capturing various EEG waveforms and degenerating data representation (Park & Kim, 2022). Motivated by these works, our paper focuses on developing methods to learn diverse, informative frequency features. In Section C.1, we provide an analysis of our proposed frequency-domain tokenizer and its impact on model performance.

## G  LLM USAGE STATEMENT

We used large language models (LLMs) solely for writing support, including grammar correction, sentence refinement, and clarity improvements. All conceptual contributions, algorithm design, code development, experiments, and analyses were conducted entirely by the authors.

