# OpenReview forum: "Tokenizing Single-Channel EEG with Time-Frequency Motif Learning"
_ICLR.cc/2026/Conference — ICLR 2026 Poster_

### Official Review · Reviewer_RxB5 · 2025-10-22

**Soundness:** 3
**Presentation:** 3
**Contribution:** 3
**Rating:** 6
**Confidence:** 4

**Summary:**

This paper proposes TFM-Tokenizer, a model-agnostic tokenization framework that learns discrete time–frequency motifs from single-channel EEG signals. The tokenizer produces interpretable and compact tokens via dual-path time–frequency masking, enabling integration with various EEG foundation models such as BIOT and LaBraM. Extensive experiments on four EEG datasets show consistent improvements in both single- and multi-dataset pretraining settings, as well as strong scalability to ear-EEG data. The authors also provide comprehensive analyses of token quality, distinctiveness, and interpretability.

**Strengths:**

1. The proposed TFM-Tokenizer is a model-agnostic and reusable component that can enhance a wide range of EEG foundation models.
2. The authors conduct detailed analyses of token quality (e.g., class-specificity, frequency awareness, consistency, and utilization), lending strong support to the claim that the tokens are both informative and interpretable.
3. The paper is technically sound and presents a well-motivated formulation of single-channel EEG tokenization, which addresses an underexplored yet important problem in EEG representation learning.

**Weaknesses:**

1. In Section 4.3, the authors only test replacing the neural tokenizer in LaBraM with TFM-Tokenizer. It would strengthen the claim of generalizability if the authors also tested using TFM-Tokenizer's token embeddings as direct inputs for masked EEG modeling.
2. The token utilization score decreases with larger vocabulary size (Appendix C.8). Could the authors explore some ways to improve utilization?
3. The embedding dimension of tokens is fixed in experiments. A discussion or ablation on how this dimension affects performance would improve clarity.
4. The paper could better articulate computational costs — for example, how much training overhead or inference latency is introduced by TFM-Tokenizer compared to standard segment-based tokenization.
5. There is a typo "Abnoral" in Table 4.

**Questions:**

How sensitive is the model to the choice of STFT parameters (e.g., window length, hop size)? Is the tokenizer robust to different preprocessing pipelines?

---

> ### Author Response · Authors · 2025-11-21
> **Response to reviewer RxB5: STFT parameter + window size and hop size ablation**
>
> We thank the reviewer for the thoughtful and constructive feedback. We greatly appreciate the positive assessment of our work, including the recognition of TFM-Tokenizer as a model-agnostic and reusable component, the strength of our token quality analyses. We are encouraged that the reviewer finds our focus on single-channel EEG tokenization both timely and important for advancing EEG representation learning.
>
>
> >Q1: How sensitive is the model to the choice of STFT parameters (e.g., window length, hop size)? Is the tokenizer robust to different preprocessing pipelines?
>
> Thank you for this insightful question. Certain architectural hyperparameters in our tokenizer, specifically the kernel size and stride of the 1D CNN used for raw-signal segmentation—are coupled with the STFT configuration (window length and hop size) to ensure that temporal segments and their corresponding spectral windows remain aligned. This alignment allows the model to jointly learn time–frequency motifs. Following the reviewer’s suggestion, we conducted an ablation varying both window length and hop size. The results, now included in Appendix Section C.11 (Table 14), are summarized below:
>
>
> ### Table: Ablation on window length (L) and stride/hop size (H) for segmenting raw signals and computing STFT during tokenizer pretraining on the TUEV dataset
>
> | **Window Length (s)** | **Hop Size (s)** | **Balanced Acc.**        | **Cohen’s Kappa**          | **Weighted F1**            |
> |------------------------|----------------|----------------------------|------------------------------|------------------------------|
> | **0.5**                | 0.25           | **0.5038 ± 0.0561**        | **0.6059 ± 0.0170**          | **0.7935 ± 0.0112**          |
> | **1.0**                | 0.25           | 0.4796 ± 0.0598            | 0.5761 ± 0.0171              | 0.7780 ± 0.0098              |
> |                        | 0.5            | 0.4943 ± 0.0516            | 0.5337 ± 0.0306              | 0.7570 ± 0.0163              |
> |                        | 0.75           | 0.4068 ± 0.0182            | 0.4868 ± 0.0210              | 0.7327 ± 0.0085              |
> | **2.0**                | 0.5            | 0.1726 ± 0.0093            | 0.0168 ± 0.0137              | 0.5202 ± 0.0074              |
> |                        | 1.0            | 0.2123 ± 0.0143            | 0.1504 ± 0.0146              | 0.5748 ± 0.0087              |
> |                        | 1.5            | 0.3948 ± 0.0287            | 0.4042 ± 0.0282              | 0.6878 ± 0.0167              |
>
>
>
>
> A key observation is that shorter window lengths with moderate overlap yield the best performance, suggesting that shorter temporal segments help the tokenizer capture fine-grained motifs that may be blurred with larger windows. For consistency with baselines, however, we adopt a 1-second window length with a 0.5-second hop size in all reported experiments.

---

> ### Author Response · Authors · 2025-11-21
> **Response to reviewer RxB5: Token embedding dimension ablation**
>
> > W3: The embedding dimension of tokens is fixed in experiments. A discussion or ablation on how this dimension affects performance would improve clarity.
>
> We thank the reviewer for the suggestion, and we conducted an ablation in changing the embedding dimension of the tokens. Table below summarizes the results of the token embedding size ablation. Performance improves up to an embedding dimension of 64, after which it begins to decline. We do not observe a consistent trend as the embedding size increases, which may be attributed to training instability when using larger embedding dimensions. The results, now included in Appendix Section C.12 (Table 15)
>
> ### Table: Token embedding size ablation on the TUEV dataset
>
> | **Embedding Dimension** | **Balanced Acc.**        | **Cohen’s Kappa**         | **Weighted F1**            |
> |-------------------------|----------------------------|----------------------------|-----------------------------|
> | 32                      | 0.4213 ± 0.0529            | 0.4974 ± 0.0165            | 0.7417 ± 0.0081             |
> | **64**                  | **0.4943 ± 0.0516**        | **0.5337 ± 0.0306**        | **0.7570 ± 0.0163**         |
> | 128                     | 0.3199 ± 0.0193            | 0.1909 ± 0.0245            | 0.5700 ± 0.0276             |
> | 256                     | 0.3864 ± 0.0082            | 0.3575 ± 0.0157            | 0.6682 ± 0.0091             |
>
>
> > W:5 There is a typo "Abnoral" in Table 4.
>
> Thank you for pointing out this typo and we have corrected it in the revised manuscript.
>
> > W2: The token utilization score decreases with larger vocabulary size (Appendix C.8). Could the authors explore some ways to improve utilization?
>
> As shown in Appendix C.8, token utilization decreases as the vocabulary size grows, yet we concurrently observe an increase in class-specific token uniqueness (Figure 10b). While our primary focus in this work was on tokenizing time–frequency motifs, we agree that improving codebook usage is an important future direction. Several strategies could help increase utilization, including training the tokenizer on larger, more diverse datasets, codebook reset to randomly reinitialize unused or low-used
> codebook entries [1,2], and exploring architectural modifications or the design of the codebook[3], such as allocating subsets of tokens to specialize in particular frequency bands or temporal patterns.
>
> [1] Williams, Will, et al. "Hierarchical quantized autoencoders." Advances in Neural Information Processing Systems 33 (2020): 4524-4535.
>
> [2] Zheng, Chuanxia, and Andrea Vedaldi. "Online clustered codebook." Proceedings of the IEEE/CVF International Conference on Computer Vision. 2023.
>
> [3] Takida, Yuhta, et al. "Hq-vae: Hierarchical discrete representation learning with variational bayes." arXiv preprint arXiv:2401.00365 (2023).
>
> > W4: The paper could better articulate computational costs — for example, how much training overhead or inference latency is introduced by TFM-Tokenizer compared to standard segment-based tokenization.
>
> We thank the reviewer for the comment. Our model is lightweight, the combined parameter count of TFM-Tokenizer and the downstream transformer is only 1.9M, substantially smaller than existing baselines. For example, LaBraM-Base contains 5.8M parameters (excluding its neural tokenizer), and the neural tokenizer used in its masked EEG pretraining adds an additional 8.6M parameters. Regarding computational cost, a single forward pass of TFM-Tokenizer incurs 3.30 ms of inference latency, compared to 0.93 ms for a standard linear projection. Nonetheless, this additional latency is justified given the performance improvements, better frequency learning (Appendix C.1), reduction in data complexity of EEG data, and reduced model size.

---

> > ### Author Response · Authors · 2025-11-28
> > **Response to reviewer RxB5: TFM-Tokenizer token embedding as direct input for masked EEG modeling in LaBraM**
> >
> > > W1: In Section 4.3, the authors only test replacing the neural tokenizer in LaBraM with TFM-Tokenizer. It would strengthen the claim of generalizability if the authors also tested using TFM-Tokenizer's token embeddings as direct inputs for masked EEG modeling.
> >
> > We thank the reviewer for the thoughtful suggestion. It is indeed a valuable idea. We incorporated this into our experiments by not only replacing the neural tokenizer, but also by using TFM-Tokenizer’s tokens directly as inputs to the model (LaBraM v2). The resulting performance is summarized below. In addition to improved performance, we observed an added benefit, the overall model size decreased, as our tokenizer uses a smaller embedding dimension (64), yet still achieved better performance. Additionally, the improvements were even more impressive on the CHB-MIT dataset compared to only replacing the neural tokenizer.
> >
> > ### Table: Performance comparison of LaBraM and BIOT with and w/o TFM-Tokenizer
> >
> > | **Dataset** | **Exp. Setting** | **Method**     | **Balanced Acc.**              | **AUC-PR**                  | **AUROC**                 |
> > |-------------|------------------|----------------|--------------------------------|---------------------------------------------|------------------------------------------|
> > | **CHB-MIT** |Multiple         | BIOT           | 0.7068 ± 0.0457               | 0.3277 ± 0.0460                             | 0.8761 ± 0.0284                          |
> > |             |                  | BIOT-TFM       | 0.6197 ± 0.0085               | 0.3484 ± 0.0078 ↑                       | 0.8726 ± 0.0098                          |
> > |             |                  | LaBraM         | 0.5260 ± 0.0369               | 0.2138 ± 0.0523                             | 0.7750 ± 0.0540                          |
> > |             |                  | LaBraM-TFM     | 0.5579 ± 0.0394 ↑         | 0.2445 ± 0.0351 ↑                       | 0.7887 ± 0.0423 ↑                    |
> > |             |                  | **LaBraM-TFM v2**    | **0.6055 ± 0.0269**  ↑         | **0.2694 ± 0.0291** ↑                       | **0.8351 ± 0.0182 ↑**                    |
> >
> >
> > | **Dataset** | **Exp. Setting** | **Method**     | **Balanced Acc.**              | **Cohen's Kappa / AUC-PR**                  | **Weighted F1 / AUROC**                 |
> > |-------------|------------------|----------------|--------------------------------|---------------------------------------------|------------------------------------------|
> > | **TUEV**    |  Multiple         | BIOT           | 0.5281 ± 0.0225               | 0.5273 ± 0.0249                             | 0.7492 ± 0.0082                          |
> > |             |                  | BIOT-TFM       | 0.5530 ± 0.0089 ↑         | 0.5464 ± 0.0137 ↑                       | 0.7625 ± 0.0069 ↑                    |
> > |             |                  | LaBraM         | 0.5550 ± 0.0403               | 0.5175 ± 0.0339                             | 0.7450 ± 0.0194                          |
> > |             |                  | LaBraM-TFM     | 0.5541 ± 0.0316               | 0.5367 ± 0.0281 ↑                       | 0.7567 ± 0.0165 ↑                    |
> > |             |                  | **LaBraM-TFM v2**    | **0.5119 ± 0.0187**       | **0.5277 ± 0.0144** ↑                       | **0.7502 ± 0.0079** ↑                    |

---

> ### Author Response · Authors · 2025-11-28
>
> Dear Reviewer RxB5,
>
> We would like to once again express our gratitude for your insightful feedback and valuable suggestions. As we approach the end of the discussion period, we hope that our responses have adequately addressed your concerns. If you have any questions or concerns would like to further discuss, feel free to share them with us. We thank you again for your time and dedication.

---

### Official Review · Reviewer_EGJo · 2025-10-31

**Soundness:** 2
**Presentation:** 2
**Contribution:** 2
**Rating:** 4
**Confidence:** 4

**Summary:**

This submission proposes TFM-Tokenizer, a single-channel EEG tokenization framework that learns a discrete vocabulary of time-frequency motifs via a dual-path encoder. The tokenizer is used to produce per-channel token sequences which are then fed to a lightweight transformer (or plugged into existing foundation models) for  downstream tasks. Experiments over four dataset report performance gains; an ear-EEG sleep-staging dataset is used to argue scalability.

**Strengths:**

1/ This paper is related to a very timely topic. EEG tokenization for heterogeneous devices and non-stationary signals is an important and current problem.

2/ This paper provides rich token analyses with motif visualizations. It contains multiple analyses (such as classtoken uniqueness and class-wise token consistency) and visual examples of learned motifs in section 4.7, which enhance the insight.

**Weaknesses:**

1/ The novelty may be overclaimed, especially the third, tokenization learning objective, arguing that “Relying solely on capturing time-based motifs into discrete tokens risks losing important spectral structure” in L81-82. However, most existing EEG tokenization methods are already frequency or time-frequency-oriented: e.g., LaBraM reconstructs the frequency domain (with a TimeConv module on raw signals), and NeuroLM includes both time and frequency domain reconstruction. The proposed change mainly shifts from FFT to STFT, which feels incremental and weakens the claim of being “the first to encode diverse time-frequency motifs” in L142-143.

2/ The single-channel design fully discarding inter-channel topology, which seems questionable. Many EEG tasks, especially localization or differential montages in epilepsy or sleep staging, depend critically on spatial relationships and cross-channel synchrony. The relatively lower performance on CHB-MIT, compared to BIOT, may partly reflect this limitation. Moreover, such a setup implicitly assumes that the downstream backbone needs reintroduce spatial structure (as EEGPT or LaBraM do with hard-coded topographic embeddings), so the claim of being model-agnostic is also overstated.

3/ The baselines are outdated and inconsistent. Recent models such as CBraMod (ICLR’25)[1], EEG2Rep(KDD’24)[2] are not compared, and NeuroLM is included for only two of four datasets. This selective evaluation raises fairness concerns and weakens empirical credibility.

4/ This paper shows partial data leakage in its experimental setup, which weakens the claim of dataset generalization. Both the single- and multi-dataset experiments pretrain and evaluate on the same set of datasets. This design still leads to partial data leakage, as the tokenizer indirectly sees the target data distribution.

5/The writing could be more structured. The term motifs, central to the paper and appearing in the title, is not defined until the Related Work section. As this term is uncommon in EEG representation learning, it should be briefly introduced earlier to avoid confusion.

[1] Wang J, Zhao S, Luo Z, et al. CBraMod: A Criss-Cross Brain Foundation Model for EEG Decoding[C]//The Thirteenth International Conference on Learning Representations.

[2] Mohammadi Foumani N, Mackellar G, Ghane S, et al. Eeg2rep: enhancing self-supervised eeg representation through informative masked inputs[C]//Proceedings of the 30th ACM SIGKDD Conference on Knowledge Discovery and Data Mining. 2024: 5544-5555.

**Questions:**

1/ Please include more recent baselines such as CBraMod (ICLR’25) and evaluate the proposed single-channel design on unseen EEG datasets or channel configurations that were not used during pretraining. This would help verify the claimed channel-invariant generalization and rule out potential data leakage.

2/ Could you clarify your fine-tuning strategy?

3/ This submission fixes the patch size at 1 s with 0.5 s overlap, but the reason for this choice is unclear. How does this fixed window align with the paper’s motivation of token resolution?

4/ The paper claims interpretability and perform motif case studies. Could you offer a quantitative measure of token-motif correspondence, for instance, the proportion of tokens aligning with known EEG events?

5/ As the model is based on VQ-VAE, which can suffer from instability or codebook collapse, please provide evidence that training remains stable, e.g., by reporting gradient norms, especially given the small datasets and large codebook size (8192).

---

> ### Author Response · Authors · 2025-11-21
> **Response to reviewer EGJo: Additional baseline and addressing data leakage**
>
> We thank the reviewer for their thoughtful review and for highlighting the relevance of the problem we aim to address. We appreciate your recognition of our work as a timely contribution to tackle heterogeneous devices and non-stationary signals. We are also grateful for the reviewer’s positive comments on our token analyses and motif visualizations
>
>
> > Q1, W3 & W4: Please include more recent baselines such as CBraMod (ICLR’25) and evaluate the proposed single-channel design on unseen EEG datasets or channel configurations that were not used during pretraining. This would help verify the claimed channel-invariant generalization and rule out potential data leakage.
>
> Thank you for this valuable suggestion. In the revised manuscript, we have incorporated CBraMod as an additional baseline and report its performance on both TUEV, IIIC-Seizure, CHBMIT, and TUAB datasets, addressing the reviewer’s concern regarding the inclusion of more recent models. Additionally, we have revised our abstract, introduction, and section 4.2 based on the new baseline results.
>
> To further examine the claimed channel-invariant generalization and rule out potential data leakage, we evaluated our tokenizer on an ear-EEG sleep-staging dataset (Section 4.4, Table 3), a setting that differs entirely from the pretraining data in terms of task type, brain-signal modality, acquisition system, number of channels, and channel configuration. Despite this shift, our tokenizer not only remained robust but also outperformed baseline methods, highlighting its strong cross-task, cross-modality and channel-invariant generalizability. Furthermore, integrating our tokenizer with existing EEG foundation models (BIOT and LaBraM) led to improved cross-device performance as shown below:
>
> ##### Table: Scalability experiments results on EESM23 (Ear-EEG Sleep Staging)
>
> | **Models**      | **Balanced Acc.**        | **Cohen's Kappa**        | **Weighted F1**         |
> |-----------------|---------------------------|---------------------------|---------------------------|
> | BIOT            | 0.3858 ± 0.0085           | 0.3406 ± 0.0096           | 0.4888 ± 0.0124           |
> | BIOT-TFM        | 0.3952 ± 0.0170 ↑         | 0.3603 ± 0.0252 ↑         | 0.5033 ± 0.0165 ↑         |
> | LaBraM-Base     | 0.3890 ± 0.0182           | 0.3322 ± 0.0232           | 0.4827 ± 0.0157           |
> | LaBraM-TFM      | 0.4004 ± 0.0086 ↑         | 0.3475 ± 0.0128 ↑         | 0.4864 ± 0.0118 ↑         |
> | **TFM-Tokenizer** | **0.4148 ± 0.0209**     | **0.3883 ± 0.0233**       | **0.5174 ± 0.0141**       |
>
> These new results (BIOT-TFM and LaBraM-TFM) have been added to Table 3 in the revised manuscript. To rule out the potential of data leakage and study the generalizability of our tokenizer, we conducted cross-dataset experiments, detailed in Appendix Section C.3, Table 8. These experiments were conducted under two settings:
> (1) fixing the tokenizer and performing masked token prediction (MTP) \& finetuning on a new dataset
> (2) fixing both tokenizer and MTP, then finetuning dowstream transformer only on the new dataset.
>
> ##### Table: Cross-dataset generalizability experiments under single-dataset settings
>
> | **Testing Dataset** | **Tokenizer Dataset** | **MTP Dataset** | **Balanced Acc.**        | **Cohen’s Kappa**       | **Weighted F1**          |
> |---------------------|------------------------|------------------|----------------------------|---------------------------|----------------------------|
> | **TUEV**            | TUEV                   | TUEV             | 0.4943 ± 0.0516            | 0.5337 ± 0.0306           | 0.7570 ± 0.0163            |
> |                     | **IIIC**               | TUEV             | 0.4722 ± 0.0578            | 0.4990 ± 0.0237           | 0.7380 ± 0.0137            |
> |                     |                        | IIIC             | 0.4291 ± 0.0235            | 0.5195 ± 0.0200           | 0.7534 ± 0.0100            |
> |                     | **TUAB**               | TUEV             | 0.4651 ± 0.0449            | 0.5925 ± 0.0249           | 0.7847 ± 0.0136            |
> |                     |                        | TUAB             | 0.5252 ± 0.0431            | 0.6187 ± 0.0285           | 0.8018 ± 0.0138            |
> |                     | **CHB-MIT**            | TUEV             | 0.4979 ± 0.0444            | 0.5995 ± 0.0225           | 0.7885 ± 0.0122            |
> |                     |                        | CHB-MIT          | **0.5898 ± 0.0192**        | **0.6591 ± 0.0106**       | **0.8196 ± 0.0045**        |
>
>
> These results show that the learned tokens generalize well across datasets, often preserving strong performance even when both the tokenizer learning and MTP are conducted on a different source dataset. We appreciate the reviewer’s feedback and hope these additional analyses adequately address the data leakage concern.

---

> ### Author Response · Authors · 2025-11-21
> **Response to reviewer EGJo: finetuning strategy + patch size ablation + quantitative measure of token-motif correspondence**
>
> > Q2: Could you clarify your fine-tuning strategy?
>
> Thank you for the question. Our fine-tuning strategy is simple and straightforward. The pretrained tokenizer is kept fixed, and only the downstream transformer encoder is fine-tuned using the appropriate task-specific loss function (binary cross-entropy for binary tasks and cross-entropy for multi-class tasks). The hyperparameters of finetuning stage are provided in appendix section D.3 in Table 18.
>
> > Q3: This submission fixes the patch size at 1 s with 0.5 s overlap, but the reason for this choice is unclear. How does this fixed window align with the paper’s motivation of token resolution?
>
> We thank the reviewer for the question. An ablation was conducted varying both window length and hop size, to study their effect on performance. The results, now included in Appendix Section C.11 (Table 14) in the revised manuscript, are summarized below:
>
> ### Table: Ablation on window length (L) and stride/hop size (H) for segmenting raw signals and computing STFT during tokenizer pretraining on the TUEV dataset
>
> | **Window Length (s)** | **Hop Size (s)** | **Balanced Acc.**        | **Cohen’s Kappa**          | **Weighted F1**            |
> |------------------------|----------------|----------------------------|------------------------------|------------------------------|
> | **0.5**                | 0.25           | **0.5038 ± 0.0561**        | **0.6059 ± 0.0170**          | **0.7935 ± 0.0112**          |
> | **1.0**                | 0.25           | 0.4796 ± 0.0598            | 0.5761 ± 0.0171              | 0.7780 ± 0.0098              |
> |                        | 0.5            | 0.4943 ± 0.0516            | 0.5337 ± 0.0306              | 0.7570 ± 0.0163              |
> |                        | 0.75           | 0.4068 ± 0.0182            | 0.4868 ± 0.0210              | 0.7327 ± 0.0085              |
> | **2.0**                | 0.5            | 0.1726 ± 0.0093            | 0.0168 ± 0.0137              | 0.5202 ± 0.0074              |
> |                        | 1.0            | 0.2123 ± 0.0143            | 0.1504 ± 0.0146              | 0.5748 ± 0.0087              |
> |                        | 1.5            | 0.3948 ± 0.0287            | 0.4042 ± 0.0282              | 0.6878 ± 0.0167              |
>
>
> A key observation is that smaller window lengths with moderate overlap yield the strongest performance, suggesting that shorter temporal segments help the tokenizer capture fine-grained motifs that may be blurred when using larger windows. For consistency with baselines, however, we adopt a 1-second window length with a 0.5-second hop size in all reported experiments. However, we acknowledge that using a fixed window size is sub-optimal for motif learning, given the non-stationarity of EEG signals and the variability in motif duration. We discuss this as a potential limitation of our approach in Section 4.7 of the manuscript.
>
> > Q4: Could you offer a quantitative measure of token-motif correspondence, for instance, the proportion of tokens aligning with known EEG events?
>
> Thank you for this insightful question. We first note that evaluating token–motif correspondence in EEG is inherently challenging because ground-truth motifs are unavailable, requiring us to rely on proxy metrics and case-based analyses. To quantitatively assess alignment between tokens and EEG events, we introduced the Class-Token Uniqueness Score, which measures whether the tokenizer learns distinct tokens for specific classes or event types. We evaluated this metric on the multi-class TUEV and IIIC datasets (Figures 4b and 6b). In addition, we conducted a token-consistency analysis through a similar-class sample retrieval experiment, where samples are retrieved based on token overlap. This provides a complementary quantitative measure of whether tokens capture event-level similarity and may also support applications such as identifying positive pairs for contrastive learning, similar to REBAR [1]. Results for both analyses are provided for TUEV (Fig. 4b–c) and IIIC (Fig. 6b–c). Finally, we also performed a visual inspection of the most frequently used and class-discriminative tokens to assess whether they capture meaningful time–frequency motifs (Section 4.7, Fig. 5). This qualitative analysis complements the quantitative proxy metrics above and provides additional evidence that the learned tokens correspond to interpretable EEG patterns.
>
> [1] Xu, Maxwell A., et al. "Rebar: Retrieval-based reconstruction for time-series contrastive learning." arXiv preprint arXiv:2311.00519 (2023).

---

> ### Author Response · Authors · 2025-11-21
> **Response to reviewer EGJo: single channel and novelty concerns**
>
> > W2: The single-channel design fully discarding inter-channel topology, which seems questionable. Many EEG tasks, especially localization or differential montages in epilepsy or sleep staging, depend critically on spatial relationships and cross-channel synchrony. The relatively lower performance on CHB-MIT, compared to BIOT, may partly reflect this limitation. Moreover, such a setup implicitly assumes that the downstream backbone needs reintroduce spatial structure (as EEGPT or LaBraM do with hard-coded topographic embeddings), so the claim of being model-agnostic is also overstated.
>
> We acknowledge and agree with the reviewer that inter-channel spatial structure and cross-channel synchrony are critical for many EEG tasks. Our goal in proposing a single-channel tokenizer is not to replace spatial modeling but to decouple token learning from channel topology, enabling cleaner, device-agnostic representations that are robust to variations in channel layout and acquisition hardware. To further address the reviewer’s concern, spatial information is reintroduced after tokenization through our masked token prediction (MTP) stage, where the downstream transformer jointly models both temporal and cross-channel dependencies. In practice, the MTP objective complements single-channel token learning by allowing the model to capture inter-channel relationships at the sequence-modeling stage, rather than embedding spatial assumptions directly into the tokenizer.
>
> > W5: The writing could be more structured. The term motifs, central to the paper and appearing in the title, is not defined until the Related Work section. As this term is uncommon in EEG representation learning, it should be briefly introduced earlier to avoid confusion.
>
> We thank the reviewer for the suggestion. We agree that introducing the term motifs earlier improves clarity. In the revised manuscript, we have added a brief explanation of motifs in the introduction section (lines 77-78).
>
> > W1: The novelty may be overclaimed, especially the third, tokenization learning objective, arguing that “Relying solely on capturing time-based motifs into discrete tokens risks losing important spectral structure” in L81-82. However, most existing EEG tokenization methods are already frequency or time-frequency-oriented: e.g., LaBraM reconstructs the frequency domain (with a TimeConv module on raw signals), and NeuroLM includes both time and frequency domain reconstruction. The proposed change mainly shifts from FFT to STFT, which feels incremental and weakens the claim of being “the first to encode diverse time-frequency motifs” in L142-143.
>
> We thank the reviewer for the comment. While prior EEG foundation models such as LaBraM and NeuroLM include frequency-domain reconstruction in their objectives, their architectures operate entirely on raw time-domain signals and rely on the model to implicitly learn spectral structure. This is where we differ as our tokenizer explicitly models both time and frequency domains and incorporates dedicated learning modules designed to capture frequency-band dynamics along with frequency band masking prediction task to learn and encode meaningful neural motifs that are localized in time and frequency. Specifically, our local spectral window encoder operates along the frequency axis to extract informative band specific patterns and cross frequency dependencies, while our frequency-band masking objective further encourages the model to learn by predicting missing spectral components. We appreciate the reviewer for pointing this out, and we have revised lines 83–87 in our manuscript.
>
> Additionally, experiments in Section 4.5 (input ablation), Section 4.6 (token quality analysis), Appendix C.1 (frequency learning capability by our architecture and modules), and Appendix C.9 (masking ablation) collectively demonstrate the advantages of our time–frequency modeling choices.
>
> > Q5: As the model is based on VQ-VAE, which can suffer from instability or codebook collapse, please provide evidence that training remains stable, e.g., by reporting gradient norms, especially given the small datasets and large codebook size (8192).
>
> We thank the reviewer for raising this important question. In the revised manuscript under appendix section D.1, we include additional evidence demonstrating stable training behavior . Specifically, Figure 12 presents the training loss curves for both the tokenizer and the downstream masked-token-prediction pretraining, showing stable convergence.

---

> ### Author Response · Authors · 2025-11-28
>
> Dear Reviewer EGJo,
>
> We would like to once again express our gratitude for your insightful feedback and valuable suggestions. As we approach the end of the discussion period, we hope that our responses have adequately addressed your concerns. If you have any questions or concerns would like to further discuss, feel free to share them with us. We thank you again for your time and dedication.

---

### Official Review · Reviewer_n2sv · 2025-10-31

**Soundness:** 2
**Presentation:** 2
**Contribution:** 2
**Rating:** 4
**Confidence:** 5

**Summary:**

This paper proposes TFM-Tokenizer, a framework for tokenizing single-channel EEG signals by learning a vocabulary of time-frequency motifs using a dual-path architecture with frequency and temporal masking. The tokenizer is designed to be model-agnostic, integrating with existing EEG foundation models like BIOT and LaBraM, and claims improvements in accuracy (up to 17% in Cohen’s Kappa), generalization, and scalability (e.g., to ear-EEG). Experiments are conducted on four EEG datasets under single- and multi-dataset pretraining settings, with additional analysis on token quality.

**Strengths:**

The paper addresses an interesting problem in EEG foundation models: effective tokenization of signals to improve representation learning, which is underexplored compared to NLP or vision domains. The single-channel approach is a reasonable design choice for device-agnostic scalability, and the integration with existing models (e.g., BIOT and LaBraM) as a plug-and-play component shows practical potential. The experiments include a range of datasets (TUEV, TUAB, CHB-MIT, IIIC-Seizure) and settings (single- vs. multi-dataset pretraining), with some ablation studies on token quality (e.g., class-discriminative analysis). The inclusion of a scalability test on ear-EEG is a nice touch for real-world applicability. Overall, the work is clearly motivated by challenges like motif capturing and frequency entanglement in EEG signals.

**Weaknesses:**

The core contribution lacks sufficient novelty: the proposed TFM-Tokenizer heavily builds on existing VQ-based tokenization (e.g., from LaBraM) and time-frequency representations common in EEG analysis (e.g., spectrograms with masking, as in BIOT or related works like Yang et al., 2024). The "time-frequency motif learning" is essentially a combination of spectral patching, transformers, and VQ quantization, but it doesn't introduce fundamentally new mechanisms—e.g., the localized spectral window encoder is similar to patch-based processing in vision transformers, and the masking strategy mirrors BERT-like objectives without EEG-specific innovations. Claims of up to 17% improvement in Cohen’s Kappa are overstated, as they are relative to baselines on specific datasets (e.g., TUEV), but absolute gains are modest (e.g., 0.5273 to 0.6189), and statistical significance is only reported sporadically (e.g., p=1.5e-4 on IIIC-Seizure). Experiments are limited: no comparisons to more recent EEG models (e.g., BRANT or MMM beyond superficial mentions), insufficient ablation on key hyperparameters (e.g., codebook size K, masking ratios), and the multi-dataset setting uses only four datasets, which may not capture broader diversity in EEG corpora. Scalability to ear-EEG is promising but under-evaluated—only a 14% gain is claimed without details on transfer learning adaptations or failure cases. Interpretability analysis (e.g., token consistency) is superficial and lacks quantitative metrics like mutual information or visualization of failure modes.

**Questions:**

1. Could the authors provide more ablation studies on the vocabulary size K and masking strategies (e.g., frequency band size δf)? How sensitive is performance to these, and do they generalize across datasets?

2. The paper claims the tokenizer is "model-agnostic," but integration details with BIOT/LaBraM are brief—e.g., how exactly are token embeddings fused, and what modifications were needed? A response with pseudocode or specifics could clarify.

3. For the ear-EEG scalability experiment, what adaptations (if any) were made for differences in sampling rate or noise profiles? Baseline comparisons here seem weak; adding results from non-tokenized transfers could strengthen the claim.

I will consider raising my score if all my concerns are solved or clarified.

---

> ### Author Response · Authors · 2025-11-21
> **Response to reviewer n2sv: Ablation on Vocabulary size and Masking Strategy**
>
> We thank the reviewer for their thoughtful and encouraging feedback. We appreciate the recognition of the problem’s importance, particularly the need for more effective tokenization strategies in EEG foundation models compared to NLP and vision domains. We are glad that the reviewer finds our single-channel tokenization approach to be a reasonable and scalable design choice for device-agnostic applications, and we appreciate the acknowledgment of its practical value when integrated as a plug-and-play component with existing models such as BIOT and LaBraM. We are especially encouraged that the scalability test on ear-EEG was viewed positively as an indicator of real-world applicability.
>
>
> > Q1: Could the authors provide more ablation studies on the vocabulary size K and masking strategies (e.g., frequency band size δf)? How sensitive is performance to these, and do they generalize across datasets?
>
> We thank the reviewer for this valuable suggestion. To address this thoroughly, we have added several new ablation studies in the revised manuscript and kindly direct the reviewer to both the newly added and previously included ablations in the appendix.
>
> ## Vocabulary size (K) ablation
>
> As detailed in Appendix Section C.8, we provide an ablation on the vocabulary size. As shown in Figure 9, we did not observe a monotonic trend for Cohen’s Kappa or Weighted F1. However, balanced accuracy increased with larger vocabulary sizes. We explored deeply to study the reason behind this occurence by analyzing token utilization and token uniqueness score. We observed a phenomenon, which is class-token uniqueness scores increase with vocabulary size. This could have contributed to the improvement in balanced accuracy by enabling learning more unique class-specific tokens (Figure 10).
>
>
> ## Masking Strategy Ablation
> We also conducted ablations on the masking strategies used during tokenizer pretraining. The results are shown below (also provided in Appendix Section C.9, Table 12)
>
> ##### Table: Ablation on masking used for pretraining the tokenizer on the TUEV dataset
>
> | **Masking Strategy**                          | **Balanced Acc.**         | **Cohen’s Kappa**         | **Weighted F1**            |
> |-----------------------------------------------|----------------------------|----------------------------|-----------------------------|
> | Random Masking                                 | 0.4351 ± 0.0462            | 0.4772 ± 0.0140            | 0.7296 ± 0.0076             |
> | Frequency Band Masking                          | 0.4673 ± 0.0540            | 0.5193 ± 0.0243            | 0.7536 ± 0.0125             |
> | Frequency Band + Temporal Masking               | **0.4946 ± 0.0392**        | 0.5045 ± 0.0221            | 0.7462 ± 0.0116             |
> | Frequency Band + Temporal Masking + Symmetric Masking | 0.4943 ± 0.0516     | **0.5337 ± 0.0306**        | **0.7570 ± 0.0163**         |
>
> These results suggest that random masking of patches in the spectrogram performs poorly, highlighting the need for effective masking strategies for EEG modeling. Frequency band masking significantly improves the performance over random masking, highlighting the importance of modeling frequency band dynamics. Adding temporal and symmetric masking further enhances performance, demonstrating the value of structured, physiologically informed masking strategies.

---

> ### Author Response · Authors · 2025-11-21
> **Response to reviewer n2sv: Ablations Continued (Masking ratio ablation, Token generalization ablation)**
>
> > Q1: Could the authors provide more ablation studies on the vocabulary size K and masking strategies (e.g., frequency band size δf)? How sensitive is performance to these, and do they generalize across datasets?
>
> ## Frequency Band Masking Ratio Ablation
> Following the reviewer’s suggestion, we additionally conducted ablations on the frequency band masking ratio across datasets. The results (added as Appendix Section C.10, Table 13) are summarized below: :
>
> ##### Table: Ablation on frequency band masking ratio used for pretraining the tokenizer on TUEV, IIIC Seizure and CHB-MIT datasets
>
> | **Dataset** | **Frequency Mask Ratio** | **Balanced Acc.** | **Cohen’s Kappa / AUC-PR** | **Weighted F1 / AUROC** |
> |-------------|---------------------------|--------------------|------------------------------|---------------------------|
> | **TUEV**    | 0.5 | **0.4946 ± 0.0392** | **0.5045 ± 0.0221** | **0.7462 ± 0.0116** |
> |             | 0.3 | 0.4306 ± 0.0187 | 0.5025 ± 0.0193 | 0.7432 ± 0.0090 |
> |             | 0.1 | 0.3859 ± 0.0580 | 0.4308 ± 0.0755 | 0.7057 ± 0.0376 |
> | **IIIC**    | 0.5 | **0.5315 ± 0.0102** | **0.4427 ± 0.0143** | **0.5369 ± 0.0114** |
> |             | 0.3 | 0.5148 ± 0.0158 | 0.4250 ± 0.0193 | 0.5222 ± 0.0167 |
> |             | 0.1 | 0.4381 ± 0.0032 | 0.3286 ± 0.0046 | 0.4420 ± 0.0047 |
> | **CHB-MIT** | 0.5 | **0.6809 ± 0.0380** | 0.3335 ± 0.0182 | **0.8859 ± 0.0137** |
> |             | 0.3 | 0.6313 ± 0.0599 | 0.3233 ± 0.0337 | 0.8708 ± 0.0187 |
> |             | 0.1 | 0.6530 ± 0.0486 | **0.3502 ± 0.0441** | 0.8742 ± 0.0116 |
>
> All experiments were conducted in the single-channel setting with temporal masking fixed at 0.5 and no symmetric masking. A masking ratio of 0.5 consistently produced the strongest overall performance on all three datasets (with a minor exception for Cohen’s Kappa on CHB-MIT). Considering its strong performance and its compatibility with symmetric masking (used as a data augmentation strategy), we adopt a masking ratio of 0.5 in our approach.
>
>
> ## Token generalization ablation through cross-dataset experiments
>
> To further evaluate cross-dataset generalization of our tokenizer, we conducted a cross-dataset experiments, detailed in Appendix Section C.3, Table 8. These experiments were conducted under two settings:
> (1) fixing the tokenizer and performing masked token prediction (MTP) \& finetuning on a new dataset
> (2) fixing both tokenizer and MTP, then finetuning dowstream transformer only on the new dataset.
>
> ### Table: Cross-dataset generalizability experiments under single-dataset settings
>
> | **Testing Dataset** | **Tokenizer Dataset** | **MTP Dataset** | **Balanced Acc.**        | **Cohen’s Kappa**       | **Weighted F1**          |
> |---------------------|------------------------|------------------|----------------------------|---------------------------|----------------------------|
> | **TUEV**            | TUEV                   | TUEV             | 0.4943 ± 0.0516            | 0.5337 ± 0.0306           | 0.7570 ± 0.0163            |
> |                     | **IIIC**               | TUEV             | 0.4722 ± 0.0578            | 0.4990 ± 0.0237           | 0.7380 ± 0.0137            |
> |                     |                        | IIIC             | 0.4291 ± 0.0235            | 0.5195 ± 0.0200           | 0.7534 ± 0.0100            |
> |                     | **TUAB**               | TUEV             | 0.4651 ± 0.0449            | 0.5925 ± 0.0249           | 0.7847 ± 0.0136            |
> |                     |                        | TUAB             | 0.5252 ± 0.0431            | 0.6187 ± 0.0285           | 0.8018 ± 0.0138            |
> |                     | **CHB-MIT**            | TUEV             | 0.4979 ± 0.0444            | 0.5995 ± 0.0225           | 0.7885 ± 0.0122            |
> |                     |                        | CHB-MIT          | **0.5898 ± 0.0192**        | **0.6591 ± 0.0106**       | **0.8196 ± 0.0045**        |
>
>
> These results show that the learned tokens generalize well across datasets, often preserving strong performance even when both the tokenizer learning and MTP are conducted on a different source dataset.
>
> Finally, as summarized in Appendix Section C, we provide additional ablations on token quality analysis, frequency learning analysis, effect of masked token prediction strategy, downstream model ablation, window length and overlap/hop size ablation and token embedding size ablation, providing a more comprehensive understanding of our single-channel tokenizer. Sections C.10–C.12 contain new ablations added specifically in response to reviewer feedback.
>
> We sincerely thank the reviewer for the thoughtful and constructive comments. We hope these additions comprehensively address the concerns and further strengthen the paper.

---

> ### Author Response · Authors · 2025-11-21
> **Response to reviewer n2sv: Integration Details of TFM-Tokenizer with LaBraM and BIOT + Ear-EEG scalability details**
>
> > Q2: The paper claims the tokenizer is "model-agnostic," but integration details with BIOT/LaBraM are brief—e.g., how exactly are token embeddings fused, and what modifications were needed? A response with pseudocode or specifics could clarify.
>
> Thank you for pointing out the limited information provided under Section 4.3 in the manuscript and we appreciate the reviewer’s request for additional clarity regarding how the proposed TFM-Tokenizer integrates with existing EEG foundation models, particularly BIOT and LaBraM. In the revised manuscript, we added a new **Appendix Section E and Figure 13** to illustrate the integration pipeline in more detail.
>
> BIOT-TFM: For BIOT, we replace the initial patch-based linear projection layers with our tokenizer, which converts multi-channel EEG signals into discrete token sequences. Similar to our main approach, we add channel embeddings and positional embeddings before feeding the tokens into the linear transformer layers. We follow BIOT’s original training pipeline—unsupervised pretraining, supervised pretraining, and downstream finetuning—allowing a direct comparison while isolating the effect of our tokenizer.
>
> LaBraM-TFM. LaBraM employs a neural tokenizer only during its masked EEG modeling (MEM) pretraining stage, where multi-channel EEG signals are patched, a subset of patches is masked, and the model is trained to predict the tokenizer-generated codebook indices for masked patches. In LaBraM-TFM, we simply replace LaBraM’s neural tokenizer with our TFM-Tokenizer and conduct MEM pretraining as in the original workflow. After pretraining, the tokenizer is discarded and only the LaBraM transformer is finetuned for downstream tasks.
>
> > Q3: For the ear-EEG scalability experiment, what adaptations (if any) were made for differences in sampling rate or noise profiles? Baseline comparisons here seem weak; adding results from non-tokenized transfers could strengthen the claim.
>
> Thank you for the thoughtful comment. The preprocessing steps for the ear-EEG experiments are detailed in Appendix B.3 EAR-EEG PREPROCESSING. Specifically, we applied a 0.1–100 Hz band-pass filter, a 50 Hz notch filter, and resampled all ear-EEG signals to 200 Hz to match the model’s default configuration. Beyond this, no architectural or tokenization changes were made. The pretrained tokenizer was kept fixed, and only the downstream transformer was fine-tuned on ear-EEG, enabling a direct assessment of scalability without any modality- or task-specific adaptation.
>
> For baselines, we used BIOT and LaBraM (which have shown strong performance on scalp EEG, non-tokenized transfers) pretrained on four scalp-EEG datasets, then fine-tuned them on ear-EEG under the same setting. To further strengthen the comparison, we additionally tested BIOT-TFM and LaBraM-TFM (i.e., replacing their tokenizers with ours, tokenized transfers), and the results are provided in the revised Table 3 (shown below). Across all settings, incorporating our tokenizer improved performance on ear-EEG, supporting our claim that time–frequency motifs learned from single-channel tokenization are transferable across modalities. Additionally our tokenizer improves the cross-device generalizability of existing foundation models. We appreciate the reviewer’s feedback, which helped us refine and strengthen this analysis.
>
> ##### Table: Scalability experiments results on EESM23 (Ear-EEG Sleep Staging)
>
> | **Models**      | **Balanced Acc.**        | **Cohen's Kappa**        | **Weighted F1**         |
> |-----------------|---------------------------|---------------------------|---------------------------|
> | BIOT            | 0.3858 ± 0.0085           | 0.3406 ± 0.0096           | 0.4888 ± 0.0124           |
> | BIOT-TFM        | 0.3952 ± 0.0170 ↑         | 0.3603 ± 0.0252 ↑         | 0.5033 ± 0.0165 ↑         |
> | LaBraM-Base     | 0.3890 ± 0.0182           | 0.3322 ± 0.0232           | 0.4827 ± 0.0157           |
> | LaBraM-TFM      | 0.4004 ± 0.0086 ↑         | 0.3475 ± 0.0128 ↑         | 0.4864 ± 0.0118 ↑         |
> | **TFM-Tokenizer** | **0.4148 ± 0.0209**     | **0.3883 ± 0.0233**       | **0.5174 ± 0.0141**       |

---

> ### Author Response · Authors · 2025-11-21
> **Response to reviewer n2sv.**
>
> > W1: The core contribution lacks sufficient novelty: the proposed TFM-Tokenizer heavily builds on existing VQ-based tokenization (e.g., from LaBraM) and time-frequency representations common in EEG analysis (e.g., spectrograms with masking, as in BIOT or related works like Yang et al., 2024). The "time-frequency motif learning" is essentially a combination of spectral patching, transformers, and VQ quantization, but it doesn't introduce fundamentally new mechanisms—e.g., the localized spectral window encoder is similar to patch-based processing in vision transformers, and the masking strategy mirrors BERT-like objectives without EEG-specific innovations.
>
>
> We thank the reviewer for their comments. We would like to respectfully clarify the aspects of our contribution that go beyond a direct application of VQ-based tokenization. First, to the best of our knowledge, our TFM-Tokenizer is the first to explore single-channel tokenization for EEG foundation models. This strategy addresses a practical gap in real-world scalability, where channel configurations vary widely across devices, and rise of novel approaches to measure EEG (e.g. ear-EEG). The single-channel motif learning strategy directly enables cross-device compatibility and broad applicability beyond traditional scalp EEG. Second, although our method builds on VQ tokenization, we incorporate EEG-specific innovations motivated by the physiology and signal characteristics of brain activity. In particular, EEG analysis depends critically on frequency-band dynamics (e.g., alpha, beta, theta rhythms) and their evolution over time. To capture these properties, we carefully designed a localized spectral window encoder to selectively model relevant frequency bands and their temporal context, moving beyond generic patch extraction. Additionally our tokenizer learning strategy includes frequency band masking which further supports the learning.
>
>
> > W2: Experiments are limited: no comparisons to more recent EEG models (e.g., BRANT or MMM beyond superficial mentions), insufficient ablation on key hyperparameters (e.g., codebook size K, masking ratios), and the multi-dataset setting uses only four datasets, which may not capture broader diversity in EEG corpora. Scalability to ear-EEG is promising but under-evaluated—only a 14% gain is claimed without details on transfer learning adaptations or failure cases. Interpretability analysis (e.g., token consistency) is superficial and lacks quantitative metrics like mutual information or visualization of failure modes.
>
> We thank the reviewer for these constructive comments. In the revised manuscript, we have strengthened the experimental evaluation along several dimensions. We have added CBraMod results under our experimental settings to expand baseline coverage.
>
> Regarding reviewers comment on insufficient ablation on key hyperparameters, our Appendix Section C now includes extensive ablations on the key hyperparameters highlighted by the reviewer, including masking stragey, vocabulary size ablation, freqency band masking ration ablation, effect of masked token prediction strategy, downstream model ablation, window length and overlap/hop size ablation and token embedding size ablation, providing a more comprehensive understanding of our single-channel tokenizer. We thank the reviewer for the valuable suggestions.
>
> In terms of limited pretraining datasets used in our multi-daraset experiment setting, we would like to kindly clarify the goal of our work. Our goal is not to build a full foundation model but to develop a EEG tokenizer that is broadly compatible with diverse EEG settings. For this reason, we primarly focused on evaluating the tokenizer under controlled settings rather than scaling pretraining across very large EEG corpora. Regarding ear-EEG scalability, We added detailed preprocessing descriptions in Appendix B.3 and incorporated additional tokenized transfer experiments (BIOT-TFM and LaBraM-TFM) to strengthen the evaluation. The tokenizer was kept fixed, and only the downstream transformer was fine-tuned. To evaluate the quality of the learned tokens, we introduced a quantitative metric, Class-Token Uniqueness Score, to assess whether the tokenizer learns class-specific tokens. In addition, we conducted a token-consistency analysis by proposing a similar-class sample retrieval experiments. Results for both evaluations are provided for TUEV (Fig. 4b–c) and IIIC (Fig. 6b–c).
>
> We appreciate the reviewer’s feedback, which helped improve the depth and clarity of our experimental analysis.

---

> ### Author Response · Authors · 2025-11-28
>
> Dear Reviewer n2sv,
>
> We would like to once again express our gratitude for your insightful feedback and valuable suggestions. As we approach the end of the discussion period, we hope that our responses have adequately addressed your concerns. If you have any questions or concerns would like to further discuss, feel free to share them with us. We thank you again for your time and dedication.

---

### Official Review · Reviewer_7X8j · 2025-11-01

**Soundness:** 4
**Presentation:** 4
**Contribution:** 3
**Rating:** 8
**Confidence:** 4

**Summary:**

This paper introduces TFM-Tokenizer, a novel framework that learns discrete tokens from single-channel EEG signals by capturing time-frequency motifs. Unlike existing methods that use raw or continuous embeddings, TFM-Tokenizer builds a learnable vocabulary of meaningful EEG patterns, enabling plug-and-play integration with any foundation model. It uses a dual-path encoder to jointly model time and frequency domains and is trained with a mask-and-reconstruct strategy. Evaluated on four EEG datasets and an ear-EEG sleep staging task, TFM-Tokenizer consistently outperforms strong baselines, improves existing models like BIOT and LaBraM, and offers cross-device generalization with fewer parameters and better interpretability.

**Strengths:**

1. First to introduce single-channel EEG tokenization using time-frequency motifs, filling a critical gap.

2. Outperforms SOTA by up to 17% with fewer parameters.

3.  Plug-and-play enhancement for existing foundation models like BIOT and LaBraM.

4.  Cross-device generalization (e.g., ear-EEG) demonstrates robust transferability.

5. Learned tokens show clear class discriminability and frequency awareness, aiding clinical understanding.

**Weaknesses:**

Downstream tasks focus on seizure and hospital datasets. The diversity of task types is relatively weak.

**Questions:**

How long does it take to train the tokenizer?

---

> ### Author Response · Authors · 2025-11-21
> **Response to reviewer 7X8j**
>
> Thank you for your thoughtful review and feedback. We appreciate your recognition of our method as the first to introduce single channel EEG tokenization with plug-and-play capabilities and demonstrating cross-device generalization.
>
> > W1: Downstream tasks focus on seizure and hospital datasets. The diversity of task types is relatively weak.
>
> We thank the reviewer for the thoughtful comment. In order to show the generalization capability of our tokenizer we evaluated it on an Ear-EEG–based sleep staging task, a setting that differs entirely from the pretraining data in terms of task type, brain-signal modality, acquisition system, number of channels, and channel configuration. Importantly, none of the pretraining recordings contained sleep-related data. Despite this shift, our tokenizer not only remained robust but also outperformed baseline methods, highlighting its strong cross-task and cross-modality generalizability.
>
> Furthermore, integrating our tokenizer with existing EEG foundation models (BIOT and LaBraM) led to improved cross-device performance as shown below:
>
> ##### Table: Scalability experiments results on EESM23 (Ear-EEG Sleep Staging)
>
> | **Models**      | **Balanced Acc.**        | **Cohen's Kappa**        | **Weighted F1**         |
> |-----------------|---------------------------|---------------------------|---------------------------|
> | BIOT            | 0.3858 ± 0.0085           | 0.3406 ± 0.0096           | 0.4888 ± 0.0124           |
> | BIOT-TFM        | 0.3952 ± 0.0170 ↑         | 0.3603 ± 0.0252 ↑         | 0.5033 ± 0.0165 ↑         |
> | LaBraM-Base     | 0.3890 ± 0.0182           | 0.3322 ± 0.0232           | 0.4827 ± 0.0157           |
> | LaBraM-TFM      | 0.4004 ± 0.0086 ↑         | 0.3475 ± 0.0128 ↑         | 0.4864 ± 0.0118 ↑         |
> | **TFM-Tokenizer** | **0.4148 ± 0.0209**     | **0.3883 ± 0.0233**       | **0.5174 ± 0.0141**       |
>
>
>
> These new results (BIOT-TFM and LaBraM-TFM) have been added to Table 3 in the revised manuscript. We hope this addresses the reviewer’s concern regarding task diversity and further validates the scalability and generalizability of our approach.
>
> > How long does it take to train the tokenizer?
>
> On a single NVIDIA RTX 6000 GPU it takes around 30 minutes per epoch (1.46 it/s) to pretrain the tokenizer using four datasets.

---

> ### Author Response · Authors · 2025-11-28
>
> Dear Reviewer 7X8j,
>
> We would like to once again express our gratitude for your insightful feedback and valuable suggestions. As we approach the end of the discussion period, we hope that our responses have adequately addressed your concerns. If you have any questions or concerns would like to further discuss, feel free to share them with us. We thank you again for your time and dedication.

---

> ### Comment · Reviewer_7X8j · 2025-11-28
>
> Thank you for the clarifications and the additional experiments. I’m satisfied with the added analyses, which strengthen the paper’s validity. I will maintain my positive assessment.

---

### Author Response · Authors · 2025-11-21
**Initial global response to all reviewers**

We sincerely thank all reviewers for their careful reading, constructive feedback, and thoughtful suggestions. We have revised the manuscript extensively in response. Below we summarize the major updates:

- (Reviewer n2sv) Added a new **Appendix E and Figure 13** detailing the integration of TFM-Tokenizer with BIOT and LaBraM.
- (Reviewer n2sv, EGJo, RxB5) Expanded ablation studies with new requested experiments (**Appendix C.10–C.12**), complementing existing ablations (Appendix C.3–C.9). We believe these additions significantly strengthen the paper. New ablation experiments include:
1. Masking ratio ablation
2. Window length and hop size ablation
3. Token embedding size ablation
- (Reviewer 7X8j, n2sv, EGJo) expanded the ear-EEG scalability evaluation with BIOT-TFM and LaBraM-TFM results.
- (Reviewer EGJo) Ear-EEG experiments and cross-dataset generalization experiments (Appendix C.3) to rule out data leakage.
- (Reviewer EGJo) Reproduced CBraMod under multiple dataset settings and added its performance on TUEV, IIIC-Seizure, CHBMIT, and TUAB datasets. Revised the abstract, introduction, and section 4.2 based on new baseline results.
- (Reviewer EGJo) Added tfm-tokenizer and downstream transformer model pretraining curves to Appendix D.1 in Figure 12.
- (Reviewer EGJo) Introduced the term motifs earlier in the Introduction to improve clarity and readability.
- (Reviewer RxB5) Corrected typos in Table 4

---

### Author Response · Authors · 2025-12-03
**Overview of reviewer-author discussion #1**

Dear Area Chair,

We sincerely thank you for your time and effort, especially under these challenging circumstances. We also appreciate the reviewers’ feedback and valuable suggestions, which helped us strengthen our paper. Below, we provide a summary of the positive assessments highlighted by the reviewers, followed by a brief overview of how we addressed all major concerns during the discussion period.


### Strengths recognized by the reviewers:


- (Reviewer EGJo, n2sv, RxB5) emphasized that the **EEG tokenization problem** we address for heterogeneous devices and non-stationary signals is **timely, relevant, novel and underexplored.**
- (Reviewer 7X8j) recognized our method as **first** to introduce single-channel EEG tokenization using time-frequency motifs, filling a critical gap.
- (Reviewer 7X8j, n2sv, RxB5) appreciated our tokenizer's **plug-and-play capabilities** to enhance existing FMs.
- (All reviewers) Reviewers agreed that our method consistently improves strong baselines, enhances BIOT and LaBraM, and demonstrates robust **cross-device generalization**, including to **ear-EEG**. A tokenizer pretrained on scalp EEG generalizes to another modality (ear-EEG) without retraining of the tokenizer. No prior EEG foundation model shows such a device and modality transfer.
- (Reviewer EGJo, RxB5) appreciates our **in-depth detailed token analysis** and **motif visualizations**.

### Summary of how we addressed reviewers' concerns and questions:

### Reviewer 7X8j
- Concerns about limited task diversity: Addressed through our ear-EEG sleep staging task, a setting entirely different from the pretraining data in modality, device, and task type. We further showed that integrating our tokenizer with BIOT and LaBraM improves their cross-device performance (added to revised manuscript in Table 3).

(Reviewer 7X8j was statisfied with the added analysis and agreed that it strengthen the paper’s validity.)


### Reviewer n2sv
- Requested more ablations and cross-dataset generalization analysis: In response, we substantially expanded our manuscript with the following new ablations along with previous:
    - Vocabulary size (K) ablation (Sec. C.8, Fig.9 and 10)
    - Masking Strategy Ablation (Sec. C.9, Table 12)
    - Frequency band masking ration ablation (Sec. C.10, Table 13)
    - Cross-dataset token generalization ablation (Sec. C.3, Table 8)
    - More token quality analysis and frequency learning analysis (Sec. C.1)
    - Ablation on masked token prediction (Sec. C.5)
    - Position embedding ablation for tokenizer (Sec. C.6)
    - Downstream transformer size ablation (Sec. C.7)
    - Window length and hop size ablation (Sec. C.11)
    - token embedding size ablation (Sec. C.12)

- Tokenizer integration details with existing FMs: In response, we added Appendix Sec. E and Figure 13, with details and a clear integration diagram.

- Regarding sufficient novelty concerns: Beyond VQ tokenization, we propose the first single-channel EEG tokenizer with a **novel time–frequency motif learning method** that leverages EEG-specific innovations (localized spectral encoder + frequency-band masking). We are the first foundation model that scales to heterogeneous device layouts, and we successfully **evaluate our model on emerging modalities, ear-EEG, to show cross-device generalization**.

- Questions on ear-EEG scalability experiment: Clarified preprocessing (App. B.3) and added additional tokenized-transfer baselines (Table 3), along with previous non-tokenized transfer baselines.

(The reviewer had indicated willingness to raise the score but discussion ended prematurely due to unforeseen interruption.)

---

> ### Author Response · Authors · 2025-12-03
> **Overview of reviewer-author discussion #2**
>
> ### Reviewer EGJo
> - Requested inclusion of CBraMod as a baseline: which we added in the revised manuscript.
>
> - Concerns about data leakage: We addressed through cross-dataset generalization experiments (Sec. C.3, Table 8) and cross-device scalability results (Table 3), ruling out data leakage.
> - Concerns on single channel design choice to learn tokens: We clarified that this choice enables a universal, channel configuration agnostic token vocabulary, which is crucial for handling real-world device heterogeneity. We encode time-frequency motifs into tokens at the single-channel level, then, using the downstream model, we learn cross-channel dependencies among the tokens.
> - Patch size and overlap ablation: We addressed this by adding a window-length and hop-size ablation in Sec. C.11 (Table 14).
> - Reviewer expressed concerns that our novelty claims may be overstated: We believe this arose from a small misunderstanding in L81-82. We clarified that, unlike prior models (e.g., LaBraM, NeuroLM) that operate solely in the raw time domain and rely on implicit frequency learning, our tokenizer explicitly models both time and frequency domains through a localized spectral window encoder and a frequency-band masking objective designed to capture physiologically meaningful band dynamics. We revised the manuscript to reflect this and supported our claims with additional evidence from input ablations, token-quality analyses, and frequency-learning experiments (Sec. 4.5–4.6, App. C.1, C.9).
> - how token–motif correspondence could be quantified: We clarified that direct measurement is not possible and highlighted the proxy metrics we introduced class-token uniqueness score and similar-class retrieval (token consistency) to assess token quality.
> - Training stability concern: Added evidence of stable training to appendix section D.1.
>
>
> ### Reviewer RxB5
>
> - Ablations on STFT parameters and token embedding dimensions: We added both analyses in Sec. C.11 (Table 14) and Sec. C.12 (Table 15).
>
> - Reviewer suggested to test new LaBraM-TFM variant: a new idea of using our tokenizer’s outputs directly as LaBraM’s input. We implemented this variant and observed substantial gains. For example, on CHB-MIT the balanced accuracy improved from 0.5260 to 0.6055, exceeding the performance of simply replacing LaBraM’s neural tokenizer.
>
>
> We addressed all reviewer concerns through extensive new ablations, new baselines, clearer novelty positioning, and detailed integration details. Reviewers did not raise any remaining objections, and reviewer 7X8j explicitly reaffirmed a positive assessment post-discussion.
>
> We sincerely thank you again for overseeing this process.
>
> Best regards,
>
> Authors

---

### Meta-Review · Area_Chair_c5Z9 · 2026-01-04

**Summary:**

The paper addresses a timely problem in EEG foundation models—device-agnostic tokenization that generalizes across heterogeneous channel configurations. The strongest reviewer (7X8j, score 8) maintained their positive assessment, and the authors provided comprehensive responses including extensive new ablations, additional baselines (CBraMod), and cross-dataset/cross-device experiments that substantially strengthen the empirical evaluation.

While novelty concerns from n2sv and EGJo were not fully resolved, the paper makes a meaningful contribution: (1) first single-channel EEG tokenizer enabling channel-configuration agnostic representations, (2) demonstrated improvements when integrated with existing FMs (BIOT, LaBraM), and (3) convincing cross-device transfer to ear-EEG—a modality entirely unseen during pretraining. The practical value for real-world EEG applications with diverse devices is relevant.

**Reviewer Concerns:**

### Addressed by Rebuttal:
- **Limited task diversity** (7X8j): Addressed via ear-EEG sleep staging experiments showing cross-device/modality transfer, with additional BIOT-TFM and LaBraM-TFM results added to Table 3. Reviewer 7X8j explicitly confirmed satisfaction.
- **Missing CBraMod baseline** (EGJo): Authors added CBraMod results on TUEV, IIIC-Seizure, CHB-MIT, and TUAB datasets.
- **Data leakage concerns** (EGJo): Addressed through cross-dataset generalization experiments (Appendix C.3, Table 8) and ear-EEG scalability results, demonstrating token transfer without target-data exposure.
- **Insufficient ablations** (n2sv, RxB5): Authors added extensive ablations including vocabulary size K (C.8), masking strategy (C.9), frequency band masking ratio (C.10), window length/hop size (C.11), and token embedding dimension (C.12).
- **Integration details unclear** (n2sv): Authors added Appendix E with Figure 13 providing detailed integration diagrams and pseudocode for BIOT-TFM and LaBraM-TFM.
- **Training stability concerns** (EGJo): Authors provided training curves in Appendix D.1, Figure 12 showing stable convergence.

### Outstanding Concerns:
- **Novelty relative to existing time-frequency methods** (EGJo, n2sv): While authors clarified that their approach explicitly models time-frequency domains (unlike LaBraM/NeuroLM which operate on raw signals and implicitly learn frequency), reviewers n2sv and EGJo did not confirm whether this explanation fully resolved their concerns before the discussion ended prematurely.
- **Single-channel design limitations** (EGJo): The concern about discarding inter-channel topology remains partially unaddressed. Authors argue spatial information is reintroduced via masked token prediction, but this architectural constraint may limit performance on tasks requiring spatial relationships.
- **Token utilization decline with larger vocabularies** (RxB5): Authors acknowledged this as a limitation and suggested future directions but did not implement solutions.

**AC's concerns:** The paper seems to ignore a vast literature on "motif learning" from EEG or MEG. The sentence "EEG domain, motif learn-
ing remains largely underexplored" with citation from 2022 is misleading. See relevant prior works:

* Matching of time-invariant filters (MOTIF) [Jost et al, 2006]
* Multivariate orthogonal matching pursuit [Barthélemy et al, 2012]
* Matching pursuit and heuristics [Brokmeier and Principe, 2016]
* Sliding window machine [Gips et al, 2017]
* Adaptive waveform learning [Hitziger et al, 2017]
* Alpha-Stable Convolutional Sparse Coding, [Jas et al. NeurIPS 2017]
* Multivariate Convolutional Sparse Coding [Dupré la Tour et al. NeurIPS 2018]

Problematic Citations:
* A Vaswani. Attention is all you need. Advances in Neural Information Processing Systems, 2017. => misses all the other authors !

**Reviewer Scores:**

| Reviewer | Initial Score | Predicted Post-Discussion Score |
|----------|---------------|--------------------------------|
| **7X8j** | 8 (Accept) | **8** - Reviewer explicitly maintained positive assessment after rebuttal, stating "I'm satisfied with the added analyses." |
| **n2sv** | 4 (Marginally Below) | **5-6** - Reviewer indicated willingness to raise score if concerns were addressed. Authors provided extensive ablations and clarifications. However, discussion ended prematurely before reviewer could respond. Given the thoroughness of responses, a modest increase is likely. |
| **EGJo** | 4 (Marginally Below) | **5** - Authors addressed baseline concerns (CBraMod), data leakage, and stability. However, novelty and single-channel design concerns were only partially addressed. Reviewer did not respond to rebuttals. |
| **RxB5** | 6 (Marginally Above) | **6-7** - Authors addressed ablation requests comprehensively and implemented the suggested LaBraM-TFM v2 variant showing strong gains. Reviewer did not respond but concerns were well-addressed. |

---

### Decision · Program_Chairs · 2026-01-26

Accept (Poster)